



# Mineral element stocks in the Yedoma domain: a first assessment in ice-rich permafrost regions

Arthur Monhonval[1], Sophie Opfergelt[1], Elisabeth Mauclet[1], Benoît Pereira[1], Aubry Vandeuren[1], Guido Grosse[2,3], Lutz Schirrmeister[2], Matthias Fuchs[2], Peter Kuhry[4], Jens Strauss[2]

[1]Earth and Life Institute, Université catholique de Louvain, Louvain-la-Neuve, Belgium
[2]Permafrost Research Section, Alfred Wegener Institute Helmholtz Centre for Polar and Marine Research, Potsdam, Germany
[3]Institute of Geosciences, University of Potsdam, Potsdam, Germany
[4]Department of Physical Geography, Stockholm University, Stockholm, Sweden

*Correspondence to*: Arthur Monhonval (arthur.monhonval@uclouvain.be)

## Abstract

With permafrost thaw, significant amounts of organic carbon (OC) previously stored in frozen deposits are unlocked and become potentially available for microbial mineralization. This is particularly the case in ice-rich regions such as the Yedoma domain. Excess ground ice degradation exposes deep sediments and their OC stocks, but also mineral elements, to biogeochemical processes. Interactions of mineral elements and OC play a crucial role for OC stabilization and the fate of OC upon thaw, and thus regulate carbon dioxide and methane emissions. In addition, some mineral elements are limiting nutrients for plant growth or microbial metabolic activity. A large ongoing effort is to quantify OC stocks and their lability in permafrost regions, but the influence of mineral elements on the fate of OC or on biogeochemical nutrient cycles has received less attention. The reason is that there is a gap of knowledge on the mineral element content in permafrost regions. Here, we use a portable X-ray fluorescence device (pXRF) to provide (i) the first large-scale Yedoma domain Mineral Concentrations Assessment (YMCA) dataset (doi:10.1594/PANGAEA.922724; Monhonval et al., in review), and (ii) estimates of mineral element stocks in never thawed (since deposition) ice-rich Yedoma permafrost and previously thawed and partly refrozen Alas deposits. The pXRF method for mineral element quantification is non-destructive and offers a complement to the classical dissolution and measurement by optical emission spectrometry (ICP-OES) in solution. This allowed a mineral element concentration (Si, Al, Fe, Ca, K, Ti, Mn, Zn, Sr and Zr) assessment on 1292 sediment samples from the Yedoma domain with lower analytical effort and affordable costs relative to the classical ICP-OES method. pXRF measured concentrations were calibrated using standard alkaline fusion and ICP-OES measurements on a subset of 144 samples ($R^2$ from 0.725 to 0.996). The results highlight that (i) the most abundant mineral element in the Yedoma domain is Si ($2739 \pm 986$ Gt) followed by Al, Fe, K, Ca, Ti, Mn, Zr, Sr, and Zn, and that (ii) Al and Fe ($598 \pm 213$ and $288 \pm 104$ Gt) are present in the same order of magnitude than OC (327-466 Gt).





## 1    Introduction

The ice-rich deposits of the Yedoma domain hold more than 25% (213 Gt) of the frozen organic carbon (OC) of the northern
circumpolar permafrost region, while covering only about 8% of its total soil area (c. 1.4 of 17.8 million km²; Schirrmeister et
al., 2013; Hugelius et al., 2014; Strauss et al., 2017). This carbon- and ice-rich region is particularly vulnerable to abrupt
thawing processes (Nitzbon et al., 2020; Turetsky et al., 2020). This is the reason why Yedoma deposits are considered as a
potential "tipping point" for future climate warming (Lenton, 2012). The key features of Yedoma deposits relative to other
permafrost deposits are (i) their ground ice properties, with 50-90% volume percent ice in Yedoma deposits (Schirrmeister et
al., 2013) and (ii) their large spatial extent and thickness, resulting in a large total volume (Strauss et al., 2017). Thawing of
ice-rich sediments has severe geomorphological consequences for the landscape (Kokelj and Jorgenson, 2013). Striking
examples are collapsing river and coastal bluffs (Fuchs et al., 2020; Günther et al., 2015; Kanevskiy et al., 2016). But a spatially
more important process is thermokarst lake formation, caused by surface subsidence due to excess ground ice melting in
Yedoma deposits leading to the formation of vast thermokarst depressions, which not only affects the superficial soil horizons
but also deep mineral horizons (Strauss et al., 2013). Thermokarst processes were highly active during the Late Glacial and
Holocene period (Walter et al., 2007) and shaped the Yedoma domain region with numerous lakes. Following drainage of
lakes, renewed permafrost aggradation could occur under new soil development in the drained thermokarst lake basins called
Alas. The Yedoma domain therefore includes Yedoma deposits never affected by thaw and frozen deposits that accumulated
after Yedoma degradation in Alas landforms (Olefeldt et al., 2016; Strauss et al., 2017).


In permafrost soils, between ~30% (Mueller et al., 2015) and ~80% (Dutta et al., 2006) of the total soil OC is associated to
minerals. It is well known that interactions between minerals and OC can have a stabilizing effect on OC (Gentsch et al., 2018;
Kaiser and Guggenberger, 2003; Kögel-Knabner et al., 2008; Lutzow et al., 2006; Wang et al., 2019). Mineral-protected OC
is less bioavailable than mineral free OC (Kleber et al., 2015; Schmidt et al., 2011), thereby contributing to the long term
carbon storage (Hemingway et al., 2019). Mineral protection of OC includes aggregation, adsorption and/or complexation or
co-precipitation processes (Kaiser and Guggenberger, 2003). These protection mechanisms involve i) clay minerals, Fe-, Al-,
Mn-oxy-(hydr)oxides, or carbonates (aggregation); ii) clay minerals and Fe-, Al-, Mn-oxy-(hydr)oxides using polyvalent
cations bridges such as $Fe^{3+}$, $Al^{3+}$, $Ca^{2+}$ or $Sr^{2+}$ (adsorption); or iii) $Fe^{3+}$, $Fe^{2+}$ and $Al^{3+}$ ions (complexation). In addition, mineral
elements (e.g., Si, Al, Fe, Ca, K, Mn, Zn, Sr) drive nutrient supply to plants or microorganisms, including algae such as
diatoms. Nutrient supply is essential for plants growth and/or microbes metabolic activity and therefore indirectly influences
C storage or degradation. It remains however unclear how mineral-OC interactions and nutrient supply will evolve upon thaw
(Opfergelt, 2020), especially in ice-rich permafrost regions. This is mainly due to a lack of knowledge about the mineral
element content in permafrost regions, relative to the well-known OC stocks in permafrost soils (Hugelius et al., 2014, 2020;
Kuhry et al., 2020; Strauss et al., 2017) and the increasing knowledge on N stocks (Fouché et al., 2020; Fuchs et al., 2018,
2019; Hugelius et al., 2020).

To improve our understanding on the potential effects mineral elements can have on OC from permafrost regions, it is essential to have better knowledge on mineral element stocks in these regions, and on the transformation of these mineral element stocks

following thawing. Ice-rich permafrost regions are particularly well suited areas to study the impact of thawing processes on the mineral elemental stocks in deposits. Indeed, 'never' (since deposition) thawed deposits can be compared with previously thawed deposits resulting from thermokarst processes of the warmer Pleistocene/Holocene transition period (13 to 9 ka BP; Morgenstern et al., 2013; Walter et al., 2007). Assessing the evolution of the mineral element stocks in these deposits will contribute to better predict the impact of the ongoing climate change on mineral element content in ice-rich permafrost regions,

with implications for the fate of OC in these deposits (Strauss et al., 2017). This is particularly relevant given that thermokarst processes are projected to spread across the Arctic and will potentially unlock additional OC stocks (Abbott and Jones, 2015; Lacelle et al., 2010; Nitzbon et al., 2020; Schneider von Deimling et al., 2015).

The aim of this study is to provide a first large-scale assessment of a climate sensitive mineral stock. We choose the Yedoma

domain and provide the Yedoma domain Mineral Concentrations Assessment (YMCA) dataset including mineral element concentrations (Si, Al, Fe, Ca, K, Ti, Mn, Zn, Sr, and Zr) and the stocks for these mineral elements in deposits from ice-rich permafrost regions. This comprises 'never' thawed since syngenetic freezing Yedoma Ice Complex deposits (in the following referred as Yedoma) and in at least once previously thawed and then refrozen drained thermokarst lake deposits (in the following referred as Alas).


## 2   Environmental settings

### 2.1   The Yedoma domain

The Yedoma domain is part of the permafrost region characterized by organic- and ice-rich deposits as well as thermokarst features. Today, the Yedoma domain covers areas in Siberia and Alaska (Figure 1) which were not covered with ice sheets

during last glacial period (110 ka – 10 ka BP; Schirrmeister et al., 2013).

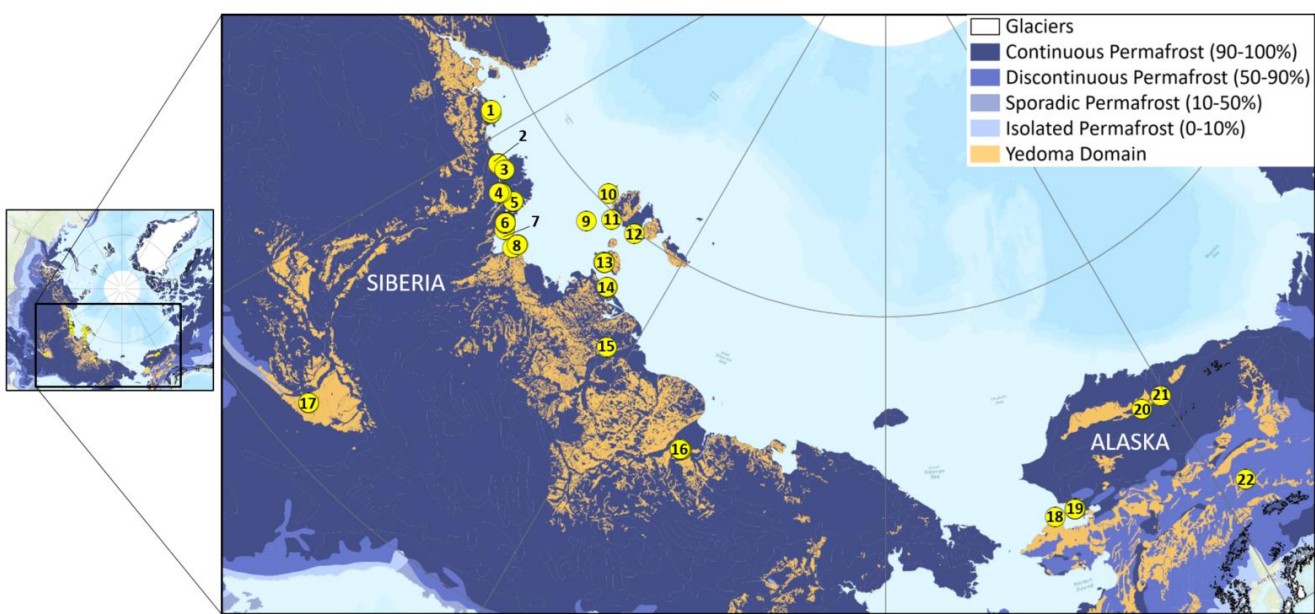

**Figure 1:** Studied permafrost sites from the Yedoma domain. 1. Cape Mamontov Klyk; 2. Nagym (Ebe Sise Island); 3. Khardang Island; 4. Kurungnakh Island; 5. Sobo Sise Island; 6. Bykovsky Peninsula; 7. Muostak Island; 8. Buor Khaya Peninsula; 9. Stolbovoy Island; 10. Belkovsky Island; 11. Kotel'ny Island; 12. Bunge Land; 13. Bol'shoy Lyakhovsky; 14. Oyogos Yar coast; 15. Kytalyk; 16. Duvanny Yar; 17. Yukechi Bassin; 18. Kitluk; 19. Baldwin Peninsula; 20. Colville; 21. Itkillik; 22. Vault Creek tunnel. Permafrost coverage layer is derived from the Circum-Arctic map of permafrost and ground-ice conditions shapefile (Brown et al., 1997). Yedoma domain coverage from Strauss et al. (2016) (Database of Ice-Rich Yedoma Permafrost (IRYP) Pangaea shapefiles).

Recent estimates indicate that the current Yedoma domain is ~1.4 million km² in extent and contains between 327 – 466 Gt OC (Strauss et al., 2017). Frozen deposits from the Yedoma domain alone store probably at least as much carbon as the tropical forest biomass (Lai, 2004). These deposits were formed by long-term continuous sedimentation and syngenetic freezing. Depositional processes are polygenetic including alluvial and aeolian deposition and re-deposition, as well as *in situ* weathering during the late Pleistocene cold stages (Schirrmeister et al., 2013). Despite evidence of homogeneity of Yedoma deposits aggradation, grain-size analyses show the large diversity of Yedoma deposits which result from multiple origins and transport of sediments, as well as (post)depositional sedimentary processes (Schirrmeister et al., 2020; Strauss et al., 2012). For millennia, continuous sedimentation lead to the accumulation of several tens of meters thick permafrost deposits with characteristic ice-wedge formation. Harsh cold late Pleistocene climate triggered the formation of freezing cracks within deposits in which water filling and refreezing created, with time, wide and up to 40 m high ice wedges. Those large volumes of ice within sediments together with the rise in temperature during the Holocene Thermal Maximum (between 11-5 ka BP depending on the region; thermal maximum reached at 7.6 - 6.6 ka BP in North Alaska and variable with time in Siberia regions; Porter and Opel, 2020) lead to a vast reshaping of the landscape with formation of thermokarst lakes and Alas basins, resulting from the drainage of former lakes (Grosse et al., 2013; Kaufman et al., 2004; Velichko et al., 2002). Alas deposits in drained thermokarst lake basins are composed of reworked Yedoma deposits as well as Holocene accumulation during sub-

aquatic and sub-aerial phases (Strauss et al., 2017; Windirsch et al., 2020). The Yedoma domain area includes 30% of unthawed

Yedoma deposits composed of homogeneous silty deposits with polygenetic origins (eolian, alluvial or colluvial deposition)

between large ice-wedges. Alas deposits have experienced thawing processes and drainage during early Holocene until more

recently and account for 56% of the Yedoma domain area. Deltaic deposits (4%) as well as lakes and rivers (10%) complete

the Yedoma domain deposits area distribution (Figure 2).

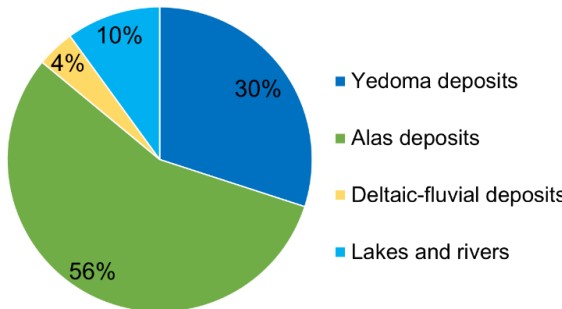

**Figure 2:** Distribution of the area coverage (percentage) in the Yedoma domain region. The total Yedoma domain area is estimated at 1 387 000 km² (based on Strauss et al., 2017).

In this study, mineral element stocks are evaluated in Yedoma and Alas deposits (Figure 3a), but the evaluation does not

include sediments underlying recent thermokarst lakes, active layer sediments or fluvial sediments (Figure 3b).


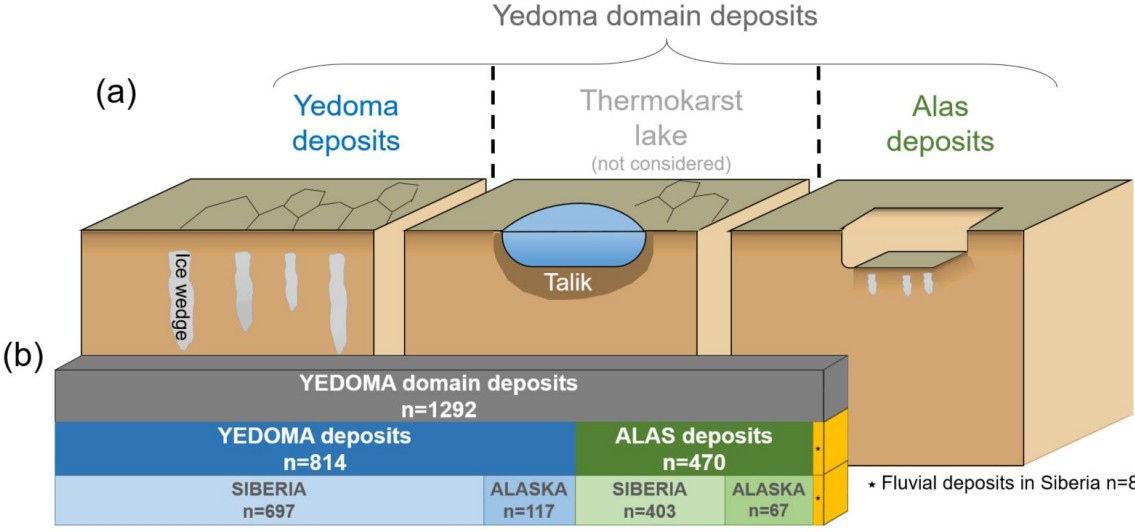

**Figure 3:** (a) Schematic view of Yedoma and Alas (drained thermokarst lake basins) deposits considered for mineral element stock calculations (modified from Nitzbon et al., 2020); sediments underlying thermokarst lakes or river floodplains are not considered. (b) Number of samples from Yedoma domain deposits included in the Yedoma domain Mineral Concentrations Assessment (YMCA) dataset (see
Sect. 4.2).



### 2.2    Sampling sites

This large-scale mineral element concentration assessment from Yedoma and Alas deposits is the result of more than 20 years of sampling in remote locations such as Siberia and Alaska. This dataset includes 22 locations from north and Interior Alaska to Laptev Sea coastal regions, Kolyma region and New Siberian Islands (Figure 1). For each location, Yedoma and Alas deposits profiles were sampled if both types of deposit were present. In total, 1292 different samples were analyzed for their mineral element concentration. Sampling strategies were specific for each sampled features. During the frozen season, samples were collected by drilling from the surface down with drilling rigs whereas during the summer season, drilling was performed below the active layer. Frozen samples from cliffs or headwall exposures were cleaned and sampled with hammers and axes. For headwall or cliff sampling, sub-profiles from different vertical exposures were included when needed to reconstruct a complete composite profile. Additional information on specific site sampling techniques can be found in reference papers cited in Table 1. Recovered samples were air- or freeze-dried before being stored and archived.



**Table 1:** Studied locations from the Yedoma domain with associated labels, the number of profiles assessed for each type of deposits and the associated reference papers. The site numbers (Site Nb) 1-17 are from Siberia, and 18-22 are from Alaska (located on the map in Figure 1). 145 The labels are used in the Yedoma domain Mineral Concentrations Assessment (YMCA) dataset (doi:10.1594/PANGAEA.922724) (see Sect. 4.2 for details). * Sobo Sise Yedoma profiles were collected from Yedoma hills but radiocarbon dating indicates that those deposits are of Holocene age.

| Site Nb | Site Name | Label | Yedoma deposits | Alas deposits | Deltaic deposits | Reference papers |
|---|---|---|---|---|---|---|
| 1 | Cape Mamontov Klyk | Mak | 2 | 1 | 0 | Schirrmeister et al., 2008, 2011 |
| 2 | Nagym (Ebe Sise Island) | Nag | 1 | 1 | 0 | Schirrmeister et al., 2003 |
| 3 | Khardang Island | Kha | 1 | 0 | 0 | Schirrmeister et al., 2007, 2011 |
| 4 | Kurungnakh Island | Bkh | 2 | 1 | 0 | Schirrmeister et al., 2003, 2008; Wetterich et al., 2008 |
| 5 | Sobo Sise Island | Sob | 2* | 1 | 1 | Fuchs et al., 2018 |
| 6 | Bykovsky Peninsula | Mkh, BYK | 5 | 3 | 0 | Andreev et al., 2002; Grosse et al., 2007; Schirrmeister et al., 2002, 2011 |
| 7 | Muostak Island | Muo | 1 | 0 | 0 | Grigoriev et al., 2003; Schirrmeister et al., 2011 |
| 8 | Buor Khaya Peninsula | Buo | 2 | 1 | 0 | Schirrmeister et al., 2016b |
| 9 | Stolbovoy Island | Sto | 3 | 1 | 0 | Grigoriev et al., 2003; Schirrmeister et al., 2003b |
| 10 | Belkovsky Island | Bel | 0 | 2 | 0 | Grigoriev et al., 2003; Schirrmeister et al., 2003b, 2011 |
| 11 | Kotel'ny Island | KyS | 1 | 0 | 0 | Grigoriev et al., 2003; Schirrmeister et al., 2003b, 2011 |
| 12 | Bunge Land | Bun | 0 | 0 | 1 | Schirrmeister et al., 2010 |
| 13 | Bol'shoy Lyakhovsky | TZ, R, L | 4 | 3 | 0 | Andreev et al., 2009; Wetterich et al., 2011, 2014 |
| 14 | Oyogos Yar coast | Oy | 1 | 1 | 0 | Opel et al., 2017; Schirrmeister et al., 2011; Wetterich et al., 2009 |
| 15 | Kytalyk | Kyt, KH | 1 | 1 | 0 | Weiss et al., 2016 |
| 16 | Duvanny Yar | DY | 5 | 1 | 0 | Strauss, 2010 |
| 17 | Yukechi Basin | Yuk-Yul | 2 | 2 | 0 | Windirsch et al., 2020 |
| 18 | Kitluk | Kit | 1 | 1 | 0 | Unpublished data ; Wetterich et al., 2012 |
| 19 | Baldwin Peninsula | Bal | 1 | 3 | 0 | Jongejans et al., 2018 |
| 20 | Colville | Col | 1 | 0 | 0 | Grosse et al., 2015; unpublished data |
| 21 | Itkillik | Itk | 1 | 0 | 0 | Kanevskiy et al., 2011 |
| 22 | Vault Creek Tunnel | FAI | 1 | 0 | 0 | Schirrmeister et al., 2016a |



## 3    Methods

### 3.1    'Classical' method for total elemental analysis: ICP-OES measurement after alkaline fusion

Inductively coupled plasma optical-emission spectrometry (ICP-OES) is a classical method to assess mineral element concentrations in solutions from the environment accurately. Solid phases such as soil samples should be first digested prior to ICP-OES analysis. Here, soils were digested by alkaline fusion. Briefly, air- or freeze-dried soil samples are carefully milled for homogenization (agate mill). We mixed a portion of the milled sample (80 mg) with Lithium metaborate and Lithium
tetraborate and heated it up to 1000 °C for 10 minutes. Then we dissolved the fusion bead in $HNO_3$ 2.2N at 80°C and stirred until complete dissolution (Chao and Sanzolone, 1992). We measured the mineral element concentrations in that solution by ICP-OES (iCAP 6500 Thermo Fisher Scientific). We assessed the loss on ignition at 1000°C, and the total element content in soils is expressed in reference to the soil dry weight at 105 °C. The analytical measurement is validated by repeated measurements on the USGS basalt reference material BHVO-2 (Wilson, 1997). To assess the precision of the method, we
conducted three repetitions on three individual samples from Yedoma and Alas deposits. For each mineral element relative standard deviations on the repetitions, expressed in % to the mean, are available in Table 2.

Out of the 1292 deposit samples retrieved from the Yedoma domain (Sect. 2.2), a subset of 144 samples has been analyzed by ICP-OES after alkaline fusion to determine their mineral element concentrations. We used this subset of analysis to calibrate pXRF-measured element concentrations (Sect. 3.2) for accurate determination of concentration values. For this first
assessment, we measured the following elements on the subset of samples (except for Zn, which was measured on 119 out of the 144 samples): Si, Al, Fe, Ca, Mg, Na, Cr, Ba, K, Ti, P, Cu, Mn, Ni, Zn, Sr, and Zr.

### 3.2    'Alternative' method for total elemental analysis: *ex situ* direct measurement by portable XRF

X-ray fluorescence (XRF) spectrometry is an elemental analysis technique with broad environmental and geologic
applications, from pollution assessments to mining industries (Ravansari et al., 2020; Rouillon and Taylor, 2016; Weindorf et al., 2014a; Young et al., 2016). In addition, there is a growing use of portable XRF for soil science (McLaren et al., 2012; Ravansari and Lemke, 2018). Portable XRF is used primarily for solid elemental analysis (soils, sediments, rocks or even plastics) but can also deal with oil chemical characterization (Weindorf et al., 2014a). XRF is based on the principle that individual atoms emit photons of a characteristic energy or wavelength upon excitation by an external X-ray energy source.
By counting the number of photons of each energy emitted from a sample, the elements present may be identified and quantified (Anzelmo and Lindsay, 1987; Appendix A). XRF-scanning results represent elements intensities in "counts per second" (cps) which are proportional to chemical concentrations in the sample but depend also on sample properties (Röhl and Abrams, 2000), ice and water content (Tjallingii et al., 2007; Weindorf et al., 2014b) and interactions between elements called "matrix effect" (Fritz et al., 2018; Weltje and Tjallingii, 2008).



*In situ* pXRF measurement often involves variability from uncontrolled environmental factors, such as water content, organic matter content or sample heterogeneity (Ravansari et al., 2020; Shand and Wendler, 2014; Weindorf et al., 2014b). To avoid such variability in water content, measurements were performed on dried samples in laboratory (*ex situ*) conditions with the handheld device in the laboratory. The particle size distribution of these deposits allows considering sample homogeneity within the fraction inferior to 2 mm (described in reference paper from Table 1). For the pXRF measurement, the dried sample

is placed on a circular plastic cap (2.5 cm diameter) provided at its base with a transparent thin film (prolene 4µm). To avoid the underestimation of the detected intensities sample thickness in the cap needs to be higher than 5 mm to 2 cm, depending on the element of interest (Ravansari et al., 2020). For a precise measurement, the sample thickness in the cap is set to >2 cm. Above 2 cm, the width is considered as "infinitely thick" for all elements. Measurements on the 1292 samples from the Yedoma domain were performed using the pXRF device *Niton xl3t Goldd+* (Thermo Fisher Scientific), which has two specific internal

calibration modes called *mining* and *soil*. Each internal calibration is dealing with different energy range and filters to scan the complete energetic spectrum from low to high-energetic fluorescence values. Both modes were used on each sample and depending on the element, the calibration with the best correlation with the classical method (Sect. 3.1) was kept for further calculations (Sect. 3.3). To standardize the analysis, total time of measurement was set to 90 seconds. We conducted the analysis in laboratory conditions, using a lead stand to protect the operator from X-rays.

In theory, the pXRF device used to generate this dataset can measure simultaneously elements of atomic mass from Mg to U. Because ambient air annihilates fluorescence photons that do not have enough energy, low atomic mass elements from Na and lighter cannot be quantified by pXRF. Note that Na quantification would be possible in controlled void conditions during analysis (Weindorf et al., 2014a). In this study, we focussed on the concentrations in 16 elements (Si, Al, Fe, Ca, Mg, Cr, Ba, K, Ti, P, Cu, Mn, Ni, Zn, Sr, Zr) by pXRF and by the classical ICP-OES method (Sect. 3.1) to allow for quality check,

calibration and correction. Some elements are at the limit of detection (LOD) for pXRF device (e.g., Cu, Ni). The LOD is reached when the sample is lacking a specific mineral element. LOD concentrations are set to 0.7 times the minimal concentration measured for this element, which is an arbitrary number but conventionally used for data statistical treatment (Reimann et al., 2008). Depending on the considered element, pXRF measured concentrations are highly precise but not always accurate (far from the true value). Trueness is achieved after correction using a regression with concentration values measured

by the classical method to avoid systematic bias (Sect. 4.1). Using a well-defined regression to correct pXRF measurements for trueness allows using the pXRF method to measure mineral element concentrations on a large number of Yedoma and Alas sample (n=1292), a valuable alternative method to assess the mineral element content on a large sample set (Figure 4).





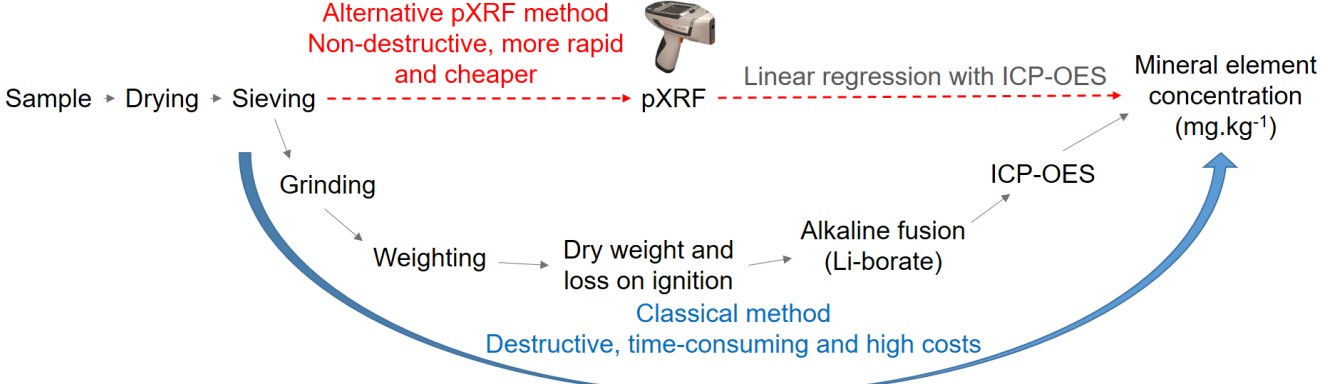

**Figure 4:** Comparison between two methods to assess mineral element content in soil samples: the classical method (large blue arrow) is destructive and time-consuming, whereas the alternative method (dotted red arrows) is non-destructive and allows a fast and reliable determination of element concentrations (correction with linear regression) on a large set of samples in a cost-effective way.

To assess the precision of the pXRF method, three to five repetitions were conducted on 20 individual samples from different locations from Yedoma and Alas deposits. Between each repetition, instrument/sample repositioning is used to mitigate the "nugget effect" due to heterogeneity in the sample (Ravansari and Lemke, 2018). Relative pooled standard deviations ($SD_{pooled}$, weighted average of standard deviations divided by weighted average of the means), expressed in %, of the repetitions for the ten elements used for stock calculation (Sect. 4.1) are available in Table 2.

**Table 2:** Precision on the element concentrations expressed relative pooled standard deviations (i.e., 2 pooled standard deviations divided by the pooled mean, expressed in %) of the pooled means element concentrations for the 10 elements considered. The precision obtained by the standard ICP-OES method can be compared with the precision of the alternative pXRF method. The values are based on a subset of Yedoma and Alas deposit samples: three repetitions of three samples for ICP-OES and on five or three repetitions of 20 samples for pXRF method.

|  |  | Si | Al | Fe | Ca | K | Ti | Mn | Zn | Sr | Zr |
|---|---|---|---|---|---|---|---|---|---|---|---|
| ICP-OES | mean (mg.kg$^{-1}$) | 315636 | 48790 | 27520 | 38221 | 12707 | 3844 | 501 | 79 | 142 | 298 |
|  | ±2SD$_{pooled}$ (%) | ± 0.6 | ± 0.9 | ± 1.0 | ± 2.0 | ± 4.0 | ± 1.4 | ± 1.7 | ± 8.6 | ± 2.1 | ± 8.4 |
| pXRF | mean (mg.kg$^{-1}$) | 319608 | 66883 | 33237 | 9766 | 19579 | 3981 | 500 | 68 | 194 | 286 |
|  | ±2SD$_{pooled}$ (%) | ± 4.3 | ± 6.1 | ± 6.9 | ± 11 | ± 5.5 | ± 7.2 | ± 17 | ± 16 | ± 8.2 | ± 10 |

### 3.3 Mean-Bootstrapping technique for mineral element stocks calculations

Calculations of mineral element stocks are based on a mean-bootstrapping technique. From the corrected mineral element concentrations obtained via linear regression of pXRF measurements (YMCA dataset; Sect. 4.2), the aim is to calculate the mineral element stock in Yedoma and Alas deposits individually, in order to estimate total mineral element budget at the





Yedoma domain scale. This technique has been used to estimate OC budget from Yedoma and Alas deposits (Strauss et al., 2013) and was improved by Jongejans and Strauss (2020). Because Yedoma domain deposits are composed of massive ice volume, stock calculations need to take into account not only the bulk density (BD) of the samples but also total thickness and total wedge-ice volume (WIV) of the deposits. In order to avoid overestimation of the mineral stocks, WIV was subtracted from the total Yedoma domain deposit volume, since the proportion of mineral elements locked inside ice wedges is negligible.

Assumptions and calculations for BD determination, WIV estimations, and deposits thickness are fully explained in Strauss et al. (2013) and are summarized here. The bootstrapping statistical method use resampled (10 000 times) observed values (mineral element concentrations, BD, WIV and deposits thickness (Appendix B)) and derive the mean afterward. The BD determination for each sample was obtained by using an inverse relationship with porosity ($\phi$) Eq. (1) (Strauss et al. 2013): $BD = (\phi - 1) \times \rho_s$ (1), whereas $\rho_s$ is the solid fraction density. We assume that pore volume, in saturated ice samples, is

directly measured with segregated ice volume. For samples where no BD determination is available, BD is inferred from the total organic carbon (TOC) content using equation Eq. (2): $BD = 1.126^{-0.061 \times TOC}$ (2) (Strauss et al., 2013). If neither BD or TOC is available, BD is fixed to 0.88 $10^3$ kg.m$^{-3}$ and 0.93 $10^3$ kg.m$^{-3}$ for Yedoma and Alas deposits, respectively, i.e., the mean BD measured in such deposits (Strauss et al., 2013) and comparable with other studies on Yedoma deposits (0.98 $10^3$ kg.m$^{-3}$; Dutta et al., 2006). The WIV estimations were performed using ice-wedge width from field measurements (Strauss et al.,

2013), ice-wedge polygon size determinations from high-resolution satellite, and additional geometrical tools (assuming ice-wedges have an isosceles triangle or rectangle shape depending on the type of ice-wedge; Strauss et al., 2013; Ulrich et al., 2014). To obtain the best approximate deposits thickness, the mean profile depths of the sampled Yedoma and Alas deposits were used following Strauss et al. (2013). Using these parameters, the overall mineral element stock is determined with Eq. (3) included into the bootstrapping calculation:


$$Mineral\ element\ stock\ [Gt] = \frac{thickness\ [m] \times coverage\ [m^2] \times BD\left[10^3\frac{kg}{m^3}\right] \times \frac{100-WIV}{100}[-] \times mineral\ concentration\left[\frac{mg_{Min\,element}}{kg_{dry\,soil}}\right]}{1,000,000,000} \quad (3)$$

Since the Yedoma domain is composed of ice-rich Yedoma deposits (30% of the area coverage) or Alas deposits (56% of the total area coverage; Figure 2) and because Yedoma and Alas deposits have different properties, from BD to ice volume content

and thickness, stocks were estimated for each deposit feature individually. Calculation for Yedoma deposits stocks are conducted as follows: the mineral element concentrations from Yedoma deposits samples (n=814) are selected (Alas deposits are not taken into account), Yedoma deposit thicknesses (n=19), and WIV properties from Yedoma deposits (n=18) are selected from Strauss et al. (2013). Total coverage is set to 410 000 km², following the last Yedoma deposits coverage estimation (Strauss et al., 2013). With all these input parameters (mineral concentration, BD, thickness, WIV and coverage), mineral

element stocks for Yedoma deposits are estimated. With a similar approach, Alas deposit mineral element stocks are estimated using the mean-bootstrapping technique. Sample properties specific to Alas deposits are selected: mineral element concentrations (n= 470) and associated BD, total thicknesses of the deposit (n=10), WIV (n=7), and the coverage is set to





780 000 km² (Strauss et al., 2013). For each bootstrapping step, the sample mineral element concentration and specific BD are paired (as they are not independent), a specific stock is estimated based on one plausible value for WIV and thickness. The

stock is then multiplied by total coverage for total mineral stock estimation to the Yedoma domain scale. From these input parameters, multiple mineral element stocks are computed, one for each bootstrapping step. Eventually, from those multiple steps (n=10 000), a mineral element stock distribution is estimated from which the mean represents the best stock estimation of the considered mineral element. The error estimates in this study represent 2 standard deviations (mean ±2σ), with the assumption of a normal distribution. Computations were performed using R software (boot package; R Core Team, 2018).

Supplementary information on input parameters (deposits thickness, WIV) are available (Appendix B).

## 4    Results

### 4.1    Linear regression for accurate mineral elements concentration measurements

Mineral element concentrations were measured with both the ICP-OES method (Sect. 3.1) and the pXRF method (Sect. 3.2) on a subset of samples (n=144) from Yedoma domain deposits. Linear regressions can be computed from this subset of data

and parameters of the regression are estimated and based on a robust linear regression (alpha=0.95; Table 3). Among the 16 mineral elements measured, 10 elements (Si, Al, Fe, Ca, K, Ti, Mn, Zn, Sr and Zr) display high correlations between pXRF and ICP-OES method with $R^2$ ranging from 0.725 (Al) to 0.996 (Ca; Table 3). Linear regression plots for these elements are presented in Figure 5. The other six elements (Mg, Ba, Cr, Cu, Ni and P) present weak ($R^2 < 0.5$) or no correlations between the two methods. According to these correlations, the mineral element stock quantification will be performed on the 10

elements reliably measured by pXRF (Si, Al, Fe, Ca, K, Ti, Mn, Zn, Sr and Zr), and the other six elements will not be discussed further in this study.

**Table 3:** Robust linear regression adjusted $R^2$ (alpha = 0.95) between concentrations measured from pXRF and ICP-OES method in Yedoma domain deposits (n=144 for all elements, except for Zn where n=119).

| Element | Si | Al | Fe | Ca | K | Ti | Mn | Zn | Sr | Zr |
|---|---|---|---|---|---|---|---|---|---|---|
| Adjusted $R^2$ | 0.835 | 0.725 | 0.949 | 0.996 | 0.941 | 0.728 | 0.875 | 0.844 | 0.965 | 0.907 |


The uncertainty on the ICP-OES measurement after alkaline fusion is always lower than pXRF measurements (Table 2). Nonetheless, the advantages of a non-destructive, rapid and cheaper method predominate for a large-scale mineral

concentration assessment given that the uncertainty (5.5% to 17%) on the pXRF measurement is satisfying to differentiate between high and low concentrations values measured in these deposits (i.e., comparing the horizontal error bar with the total range of values represented on the X-axis in Figure 5).



Earth System
Science
Data

In this dataset, the risk of overestimating the element concentration by pXRF in organic-rich samples (Shand and Wendler, 2014) is limited. Among the 1292 Yedoma and Alas samples used in this study to build the YMCA dataset (Sect. 4.2), less

than 0.5% of the samples are characterized by TOC content similar to or higher than 40 wt% (Table 4). Overestimation of the concentration was only observed for a single sample for Fe (Figure 5c) and for three samples for Ca (Figure 5d) out of the 144 samples from the subset (for samples with TOC content > 40 wt%), and not observed for Si, Al, K, Ti, Mn, Zn, Sr and Zr. Since the points corresponding to these organic-rich samples are excluded from the robust linear regressions (Figure 5), it can be considered that the matrix effect is not a source of significant bias to correct pXRF measurements of element concentrations

using these regressions. Thus, the pXRF method can be applied for this first large-scale mineral element assessment of Yedoma domain deposits.

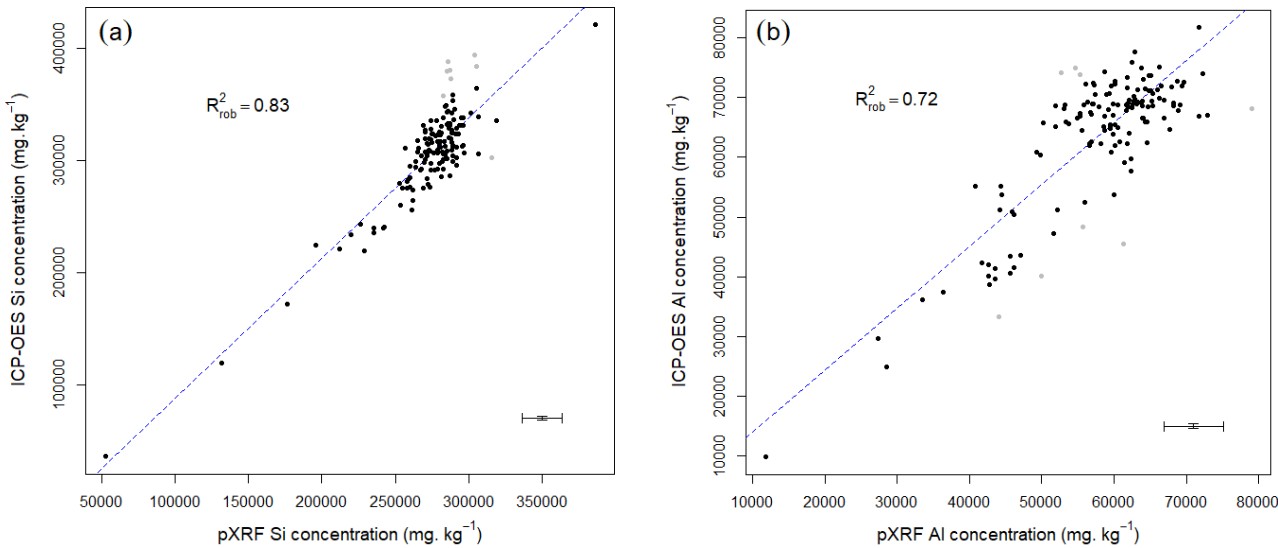

**Figure 5:** Robust linear regressions (blue line) between element concentrations obtained by the ICP-OES method as a function of the
concentrations obtained by the pXRF method. The data are presented for the 10 elements considered: Si (a), Al (b), Fe (c), Ca (d), K (e), Ti (f), Mn (g), Zn (h), Sr (i), and Zr (j) with n=144 for all elements, except for Zn where n=119. Grey points (which include organic-rich samples; TOC > 40 wt%) are excluded from the robust linear regression. Errors bar (±2 pooled σ) are calculated based on three repetitions of three samples for ICP-OES method and on three to five repetitions of 20 samples for pXRF method.






**Figure 5 (Suite)**



**Figure 5 (Suite)**


## 4.2 Yedoma domain Mineral Concentrations Assessment (YMCA) dataset

Robust linear regression equations with reliable R² (Table 3) were applied for each pXRF-measured concentration to correct concentrations values for trueness. The resulting YMCA dataset compile 1292 samples for the following elements: Si, Al, Fe, Ca, K, Ti, Mn, Zn, Sr and Zr.



In addition to the corrected mineral element concentrations, the YMCA dataset combines all the relevant existing information on these 1292 samples. Site and samples properties are integrated from AWI (Alfred Wegener Institute) Potsdam and Stockholm University data. We associated the lithology of the underlying bedrock, and the type of unconsolidated sediments at the surface characterizing each site from a spatial join using Arc Map 10.4. Specifically, coordinates from each site are joined based on their spatial location to Global Lithology Map (GLiM; Hartmann and Moosdorf, 2012) and Global

Unconsolidated Sediments Map (GUM; Börker et al., 2018). Typically, we included the lithology of the underlying bedrock to evaluate its potential control on the mineral element concentrations in the deposits.

Practically, the columns from this dataset available at doi:10.1594/PANGAEA.922724 (Monhonval et al., in review) are organized as follow: (i) all site and sample properties (sample ID, type of deposit, site location, number of profile, GPS coordinates, country, GLiM-lithology, GUM-unconsolidated sediment type, GUM-age, sample depth below surface level

(b.s.l) or height above sea/river level (a.s.l), sediment characteristics, BD, gravimetric and absolute ice content, TOC values); (ii) corrected pXRF concentrations based on linear regressions from Sect. 4.1. Table 4 summarizes corrected mineral element concentration values as well as BD, TOC and ice content from the whole Yedoma domain region, with no differentiation between Yedoma and Alas deposits samples (n=1292).






**Table 4:** Statistical summary of YMCA dataset (n=1292) for bulk density (BD), total organic carbon content (TOC), absolute ice content, and corrected mineral element concentrations (Si, Al, Fe, Ca, K, Ti, Mn, Zn, Sr and Zr) in Yedoma domain deposit samples (grouping Yedoma and Alas deposits together). Columns stand for N (number of data); Missing (missing data); MIN (minimum value reported); Q (quantiles); Q1 (first quantile); Q3 (third quantile); MEDIAN; MEAN; MAX (maximum value reported); SD (standard deviation); MAD (median absolute deviation standardised to conform with the normal distribution); IQR (interquartile range standardised to conform with the normal distribution); CV (coefficient of variation (%)); CVR (robust coefficient of variation (%)). For Si, excessive organic rich samples may result in lower concentration values and consequently in negative concentration value after correction. Only positive values have been considered for stock calculations.

| | Units | N | Missing | MIN | Q_0.05 | Q1 | MEDIAN | MEAN-log | MEAN | Q3 | Q_0.95 | MAX | SD | MAD | IQR | CV | CVR |
|---|---|---|---|---|---|---|---|---|---|---|---|---|---|---|---|---|---|
| BD | g.cm$^{-3}$ | 1292 | 0 | 0.0379 | 0.721 | 0.943 | 0.988 | 1.076 | 1.143 | 1.322 | 2.055 | 2.429 | 0.397 | 0.1083 | 0.2814 | 34.74 | 10.96 |
| TOC | wt% | 1148 | 144 | 0.07 | 0.1 | 1.135 | 1.953 | 1.878 | 3.783 | 3.76 | 13.34 | 44.36 | 6.015 | 1.613 | 1.946 | 159 | 82.58 |
| Abs. ice content | % | 703 | 589 | 0 | 17.76 | 28.83 | 37.5 | 36.66 | 38.94 | 47.8 | 64.69 | 103.9 | 14.99 | 14.08 | 14.06 | 38.48 | 37.56 |
| Si | mg.kg$^{-1}$ | 1292 | 0 | -19310 | 241400 | 284000 | 301000 | 290400 | 295200 | 316900 | 335700 | 444700 | 40750 | 24170 | 24330 | 13.8 | 8.031 |
| Al | mg.kg$^{-1}$ | 1292 | 0 | 6480 | 48630 | 61210 | 66460 | 63700 | 64820 | 70600 | 77100 | 101000 | 9978 | 6914 | 6957 | 15.39 | 10.4 |
| Fe | mg.kg$^{-1}$ | 1292 | 0 | 9591 | 18770 | 28270 | 31680 | 30970 | 32140 | 34470 | 45810 | 111000 | 9234 | 4586 | 4600 | 28.73 | 14.48 |
| Ca | mg.kg$^{-1}$ | 1292 | 0 | 2086 | 5316 | 7972 | 10380 | 11490 | 13910 | 15520 | 32910 | 102000 | 11540 | 4667 | 5594 | 82.94 | 44.97 |
| K | mg.kg$^{-1}$ | 1292 | 0 | 2097 | 11820 | 17680 | 19290 | 18300 | 18720 | 20640 | 22580 | 30380 | 3270 | 2192 | 2194 | 17.47 | 11.36 |
| Ti | mg.kg$^{-1}$ | 1292 | 0 | 100.2 | 2112 | 3715 | 4171 | 3867 | 4015 | 4480 | 5081 | 10970 | 940.1 | 536.9 | 567.5 | 23.41 | 12.87 |
| Mn | mg.kg$^{-1}$ | 1292 | 0 | 118.6 | 275.8 | 409 | 498 | 490 | 569.9 | 568.2 | 775.1 | 48690 | 1395 | 118.8 | 118 | 244.8 | 23.86 |
| Zn | mg.kg$^{-1}$ | 1292 | 0 | 22.1 | 29.88 | 58.45 | 70.86 | 67.01 | 71.28 | 83.37 | 103.8 | 686.8 | 28.81 | 18.47 | 18.48 | 40.41 | 26.07 |
| Sr | mg.kg$^{-1}$ | 1292 | 0 | 32.82 | 96.03 | 158.1 | 186 | 185 | 196.2 | 242.5 | 308.1 | 392.5 | 63.53 | 59.81 | 62.57 | 32.38 | 32.16 |
| Zr | mg.kg$^{-1}$ | 1292 | 0 | 43.74 | 158.6 | 242.8 | 278.5 | 269.4 | 280.1 | 314 | 399.8 | 711 | 73.48 | 53.03 | 52.8 | 26.23 | 19.04 |





### 4.3 Mineral element stocks in the Yedoma domain

The quantification of mineral element stocks is a major step to assess the distribution of mineral elements in the Yedoma domain. We performed mineral elements stock estimations with the bootstrapping technique (Sect. 3.3).

For example, the mean Fe stock from Yedoma deposits (n=814) is $147 \pm 43$ Gt ($\pm 2\sigma$) using the theoretical normal distribution
(mean-bootstrapping technique; Figure 6).

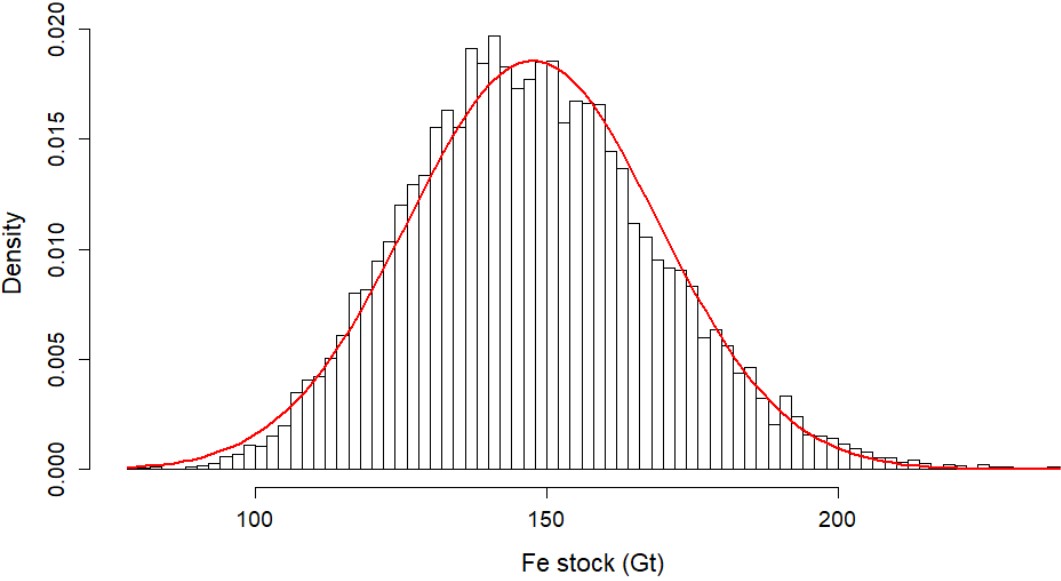

**Figure 6:** Distribution of the iron stock (Gt) in Yedoma deposits (n=814). The histogram results from the bootstrapping technique (n=10 000; Appendix B). The theoretical normal distribution is shown in red.

Based on the YMCA dataset, the bootstrapping technique is applied to calculate the stocks in Yedoma deposits for the 10 elements (Si, Al, Fe, Ca, K, Ti, Mn, Zn, Sr, and Zr), basing on the same input parameters for BD, thickness, WIV and coverage. The estimate mean and standard deviation derive the best estimation of the Yedoma deposit stocks for each specific mineral element. With a similar approach, mineral element stocks (Si, Al, Fe, Ca, K, Ti, Mn, Zn, Sr, and Zr) are estimated in Alas deposits. The element stocks in the Yedoma domain are obtained by adding element stocks in Yedoma and Alas deposits
(Table 5). The results highlight that the mineral element with the largest stock in the Yedoma domain is Si ($2739 \pm 986$ Gt) followed by Al, Fe, K, Ca, Ti, Mn, Zr, Sr, and Zn (Figure 7). In comparison, the estimated OC stock in the Yedoma domain reaches around 324-466 Gt (Strauss et al., 2017; Figure 7).





**Table 5:** Mineral elements stock (Gt) summary in Yedoma, Alas and Yedoma domain deposits (n=814, 470 and 1284, respectively). Mean and associated 2 standard deviations (σ; absolute value) are provided. Total stocks are the addition from the stocks in Yedoma and Alas deposits. Note, fluvial /marine deposits stocks were not estimated because of lack of data and spatial coverage (n=8).

| | Stock (Gt) | | | | | |
| | Yedoma deposits | | Alas deposits | | Total Yedoma domain | |
| Element | mean | ± 2σ | mean | ± 2σ | mean | ± 2σ |
|---|---|---|---|---|---|---|
| Si | 1413 | 408 | 1327 | 578 | 2739 | 986 |
| Al | 309 | 88.9 | 289 | 124 | 598 | 213 |
| Fe | 147 | 42.6 | 140 | 60.9 | 288 | 104 |
| Ca | 70.8 | 20.9 | 51.7 | 23.2 | 123 | 44.1 |
| K | 90.0 | 25.7 | 84.0 | 36.8 | 174 | 62.4 |
| Ti | 19.0 | 5.45 | 17.5 | 7.68 | 36.5 | 13.1 |
| Mn | 2.57 | 0.852 | 2.34 | 1.02 | 4.90 | 1.87 |
| Zn | 0.327 | 0.0941 | 0.311 | 0.137 | 0.638 | 0.231 |
| Sr | 0.963 | 0.278 | 0.935 | 0.409 | 1.90 | 0.687 |
| Zr | 1.36 | 0.393 | 1.25 | 0.543 | 2.61 | 0.936 |

Overall, Yedoma and Alas deposits (Table 5) represent 86% of the total Yedoma domain area (Figure 2): this means that mineral element stocks are based on 86% of the deposits. The remaining surfaces of the Yedoma domain correspond to deltaic deposits (4%) or lakes and rivers (10%). Element stocks in deltaic deposits have not been estimated in this study due to the scarce number of available mineral concentration data in deltaic deposits (0.6% of the YMCA dataset) and the lack of empirical estimations of their thickness, or their ice volume (even if ice volume is probably negligible in these deposits). Waterbodies,

lakes and rivers are not considered in this assessment focusing on the mineral element content in surface solid materials vulnerable to thaw or already thawed in the past. Sediments underlying lakes and rivers are therefore outside the scope of this evaluation and would only become relevant if drainage occurs. Absolute stock estimates (in Gt) allow direct comparison between Yedoma and Alas deposits, despite their different thickness, WIV and coverage. Stock estimation (kg.m$^{-3}$) are available (Appendix D) in which WIV have been neglected: this estimation is the product of mineral element concentrations

and BD using bootstrapping technique.

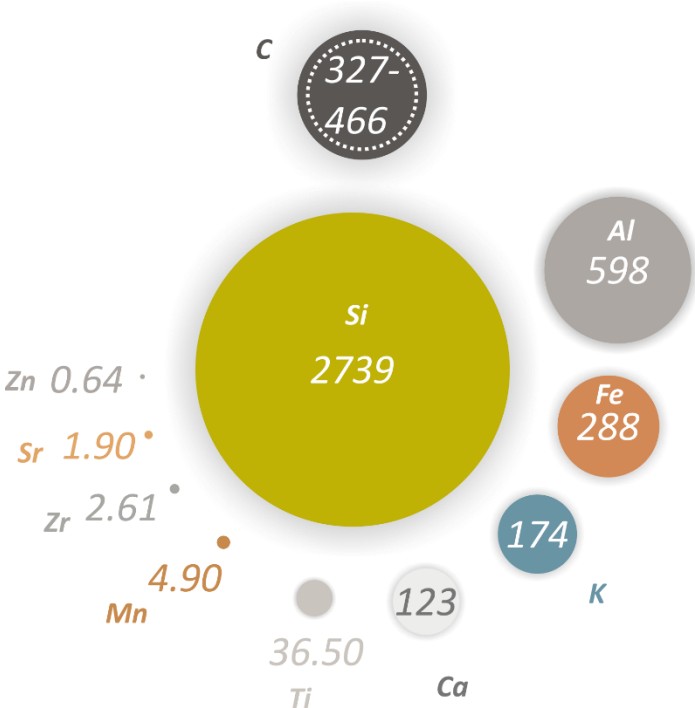

**Figure 7**: Mineral element stocks (in Gt) in the Yedoma domain deposits (including Yedoma and Alas deposits) for Si, Al, Fe, K, Ca, Ti, Mn, Zr, Sr, and Zn. The surface area of the circles is proportional to the stock of the element considered. The organic carbon stock in the Yedoma domain (Strauss et al., 2017) is provided for comparison (the dotted line represents the lower range of the estimate, i.e., 327 Gt).

## 5 Discussion

### 5.1 Advantages and limitations of the ICP-OES and pXRF method for mineral element concentration measurements

Although considered as a classical method for total element analysis, the main drawbacks of the ICP-OES technique after alkaline fusion are to be a destructive, multiple steps and time consuming (involving more risks of error), resulting in high costs analytical protocol when dealing with thousands of samples. Indeed, this protocol includes: (i) the homogenization of samples before weighting, a critical step to avoid heterogeneities or nugget effect, (ii) the determination of dry weight for each sample to correct for sample's moisture, (iii) the need to assess loss on ignition to close the budget of the sample oxide content (Figure 4). For these reasons, we have investigated for a suitable alternative method.

The pXRF device ensures a non-destructive, rapid and cheaper method for mineral element quantification compared to the classical ICP-OES measurements after alkaline fusion when dealing with large-scale assessments involving thousands of samples. The pXRF method allows the determination of mineral elements concentrations as well as first stock calculations for Si, Al, Fe, K, Ca, Ti, Mn, Zr, Sr, and Zn (n=1292; Figure 7) with minimal analytical effort relative to the classical wet chemistry method. Drawback related to inaccurate pXRF measurements is rectified by correcting those values with accurate values





obtained with the classical ICP-OES method after alkaline fusion (n=144). From this YMCA dataset, mineral element stocks
were estimated using a mean-bootstrapping technique previously applied to estimate the OC budget in the Yedoma domain
(Strauss et al., 2013). However, some important mineral elements could not be assessed because of poor correlation with
classical ICP-OES method (Mg, P, Cu, Ni, Ba) or because of their low atomic masses which make XRF-measurement
impossible (N, Na, S).

**5.2     Implications for mineral element mobility upon thermokarst processes**

The YMCA dataset allows investigating the mineral element behavior upon thaw. Mineral element concentrations of some
soluble elements, such as Ca, could be influenced by thermokarst processes. Indeed, Alas formation history includes lake
formation and drainage, and the dynamism of such formation over the past thousands of years may lead to leaching processes
of soluble elements, a commonly observed process in soils (Stumm and Morgan, 1995) and cryogenic soils (Ping et al., 2005).
As shown on a density plot (Figure 8), sediments from Alas deposits are characterized by lower Ca concentrations compared
to Yedoma deposits. We made this observation between Alas and Yedoma in Alaska, and between Alas and Yedoma in Siberia
(Figure 8). The higher Ca concentrations in Yedoma and Alas deposits from Alaska relative to the deposits from Siberia can
be explained by the local lithology, i.e., carbonate rocks from the northern Brooks Range contributing to the deposits in Alaska
(Till et al., 2008). The lithology of the underlying bedrock inferred from the GLiM map is similar between Yedoma and Alas
deposits (Appendix C). This can likely be explained by the fact that the deposits from the Yedoma domain mainly originates
from Quaternary deposits overlying the bedrock (Grosse et al., 2007; Strauss et al., 2017). Therefore, mineral element
concentrations are more influenced by the mixing of the unconsolidated sediments contributing to the Quaternary deposits
rather than by the lithology of the underlying bedrock. The lithological similarity between Yedoma and Alas deposits can be
also explained by the fact that Alas deposits are dominated by reworked sediments from former Yedoma deposits
(Schirrmeister et al., 2020). Therefore, Ca depletion in Alas deposits relative to Yedoma deposits from one region is not
lithology dependent but potentially results from leaching processes of soluble elements such as Ca during former thawing
periods.





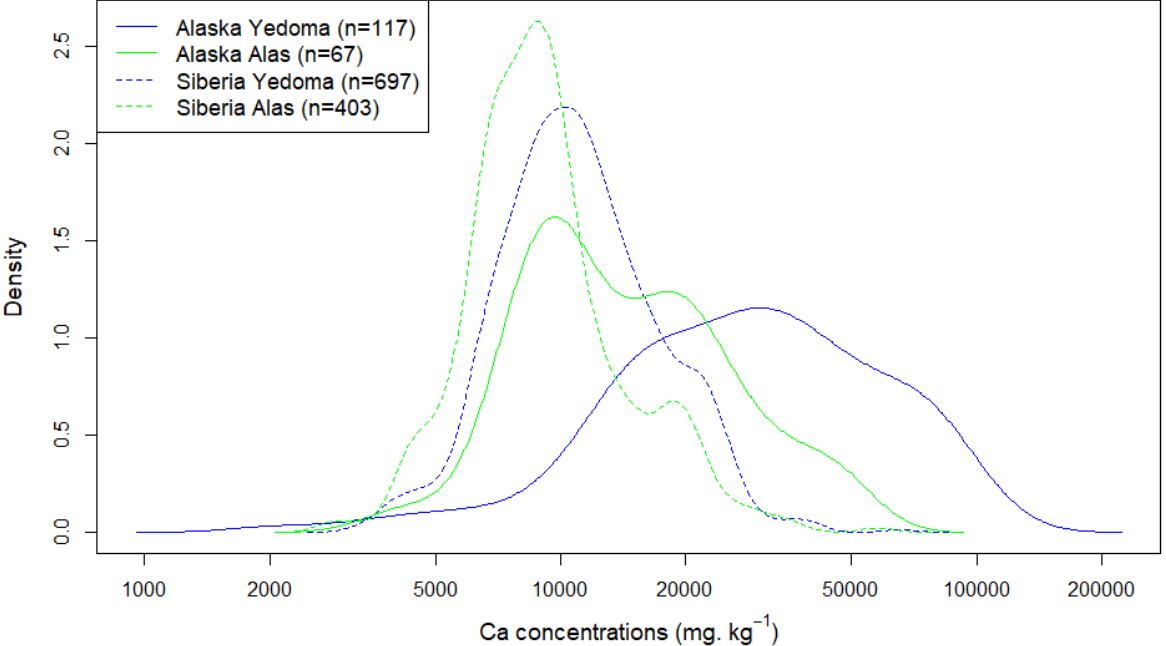

**Figure 8:** Empirical density histogram of calcium concentrations (mg.kg⁻¹) in Yedoma domain deposits in Alaska (full line) and Siberia (dashed line) for Yedoma (blue line) and Alas (green line) deposits. The X-axis is a log scale.

This observation is based on the YMCA dataset comprising more than a thousand samples supporting that Ca solubility and leaching mechanisms following Holocene thermokarst processes is a suitable hypothesis to explain lower Ca concentration in Alas deposits relative to Yedoma deposits. This highlights the potential of our study and the YMCA dataset to investigate dynamic processes controlling mineral element concentrations in thawing environments.



## 5.3 Implications of mineral elements release upon permafrost thaw

### 5.3.1 Implications for organic carbon stabilization/degradation

The evolution of interactions between mineral elements and OC upon thaw is of major concern to predict OC stabilization/degradation patterns in the context of permafrost thaw. Changing conditions for mineral protection of OC upon thawing is likely to influence OM degradation (Herndon et al., 2017; Kögel-Knabner et al., 2010; Opfergelt, 2020). Indeed, mineral elements play a key role for OC microbial degradation. This was illustrated by incubation experiments highlighting

the importance of organo-mineral interactions for soil OC sequestration (Gentsch et al., 2015), or by the inhibition effect of some mineral elements (e.g., Fe, Mn) on methane production (Beal et al., 2009; Herndon et al., 2015; Lovley and Phillips, 1987; Sowers et al., 2018). The OC can be stabilized via i) spatial inaccessibility of OM against decomposers organisms due to occlusion in aggregates; ii) organo-mineral associations with Fe-, Al-, Mn-oxides or clay minerals using polyvalent cations bridging between OC and mineral surfaces (e.g., $Ca^{2+}$, $Sr^{2+}$ in neutral and alkaline soils or $Fe^{3+}$ and $Al^{3+}$ in acid soils); or iii)

organo-metallic complexes involving $Fe^{3+}$, $Fe^{2+}$ and $Al^{3+}$ ions (Lutzow et al., 2006). Up to 80% of the total soil C can be stored within mineral horizons in the first meter of tundra soils of Arctic Siberia (Gundelwein et al., 2007). Providing a first assessment of the mineral element content in the Yedoma domain deposits (the YMCA dataset) identifies the mineral elements (e.g. Al, Fe, Ca, Mn and Sr) potentially available to interact with OC in permafrost landscapes. Mineral elements from permafrost could contribute to modulate the permafrost carbon feedback through the evolution of mineral-OC interactions

upon thaw.

### 5.3.2 Implications for the supply of nutrients

Other processes through which mineral elements from the permafrost can affect the carbon cycle and the permafrost carbon feedback is by indirect pathways such as nutrient supply for terrestrial or aquatic Arctic ecosystems. Macro- (e.g., K, Ca) and

micro-nutrients (e.g., Fe, Mn, Zn) are required for plant nutrition and also regulates other vital processes for plants and microorganisms growth and metabolic activity (DalCorso et al., 2014). Briefly, K regulates vital processes, such as photosynthesis, water and nutrient transportation, or protein synthesis (Marschner, 2012). Ca is a major second messenger in plant signal transduction, mediating stress- and developmental processes (Liese and Romeis, 2013). Micro-elements are required as cofactor for some essential proteins (Fe; Morgan and Connolly, 2013), play an essential role as a photosynthetic

function or in metallo-protein conformation (Mn; Yang et al., 2008) or can have enzymatic functions (Zn; Lindsay, 1972). Some non-essential elements, such as Si and Al, can stimulate plant growth by playing with abiotic-biotic stress resistance and symbiosis (DalCorso et al., 2014; Richmond and Sussman, 2003). In aquatic ecosystems, silicon is a limiting nutrient for diatoms and other siliceous organisms, thereby controlling diatom abundance and community structure in the ocean, and as a result, food web and $CO_2$ uptake by photosynthesis (Smetacek, 1999; Yool and Tyrrell, 2003). Assessing the concentration of

elements considered as immobile such as Ti or Zr can be useful to evaluate the advance of weathering in a soil relative to its



parent material, and thereby the soil mineral reserve (Hodson, 2002; Jiang et al., 2018; Kurtz et al., 2000): the leaching of more mobile cations such as Ca or K can be estimated by comparing ratios between mobile and immobile elements. Other important nutrients (e.g., P, N, S, Mg) could not be estimated in this study (Sect. 5.1). Therefore, the YMCA dataset is far from a comprehensive dataset but remains a first step needed for ice-rich permafrost mineral assessment.

**6    Data availability**

The Yedoma domain Mineral Concentrations Assessment (YMCA) dataset is available online at https://doi.pangaea.de/10.1594/PANGAEA.922724 (Monhonval et al., in review).

**7    Conclusions**

This study provides the first mineral element inventory of permafrost deposits focusing on the ice-rich Yedoma region, i.e.,
never thawed Yedoma deposits and previously thawed Alas deposits. Based on the YMCA dataset, the stocks of 10 mineral elements in Yedoma domain deposits have been quantified. Mineral elements stocks are shown to be in the same order of magnitude for Al and Fe than for OC, and to decrease from Si, Al, Fe, K, Ca, Ti, Mn, Zr, Sr, to Zn. Among the 10 investigated elements, Si has the highest stock with about 2700 Gt. This dataset allows tracking dynamic processes controlling mineral element concentrations in thawing environments, as illustrated by lower Ca concentration in Alas deposits relative to Yedoma
deposits highlighting potential Ca leaching upon thawing. This dataset also provides a vertical distribution of 10 important mineral elements in 75 different profiles from the Yedoma domain. In permafrost soils, between 30 and 80% of OC is considered to be mineral-protected, i.e., involving interactions between OC and mineral constituents or mineral elements. Providing the YMCA dataset is contributing to improve the knowledge on the mineral side of permafrost, i.e., a necessary step to better understand the evolution of mineral-OC interactions upon permafrost thaw. The YMCA dataset is particularly relevant
given the increasing occurrence of abrupt thaw in ice-rich permafrost regions. Abrupt thaw exposes deep mineral horizons, and thereby a stock of mineral elements potentially available to interact, directly or indirectly, with OC and influence the fate of OC upon thaw.




# 8     Appendices

500          **Appendix A**

**Fig. A1:** Principles of the X-ray fluorescence methodology. An X-ray source emitted by the portable XRF device (here, *Niton xl3t Goldd+,* Thermo Fisher Scientific) excites and ejects an electron from a specific atom. An outer electron fills vacancy to stabilize the overall atom and the electron translocation process implies an energy loss (fluorescence). This energy, specific to each atom, is detected, amplified and processed by the pXRF device. The processing converts photon energy of a specific wavelength to counts per seconds to concentration of a
specific element.

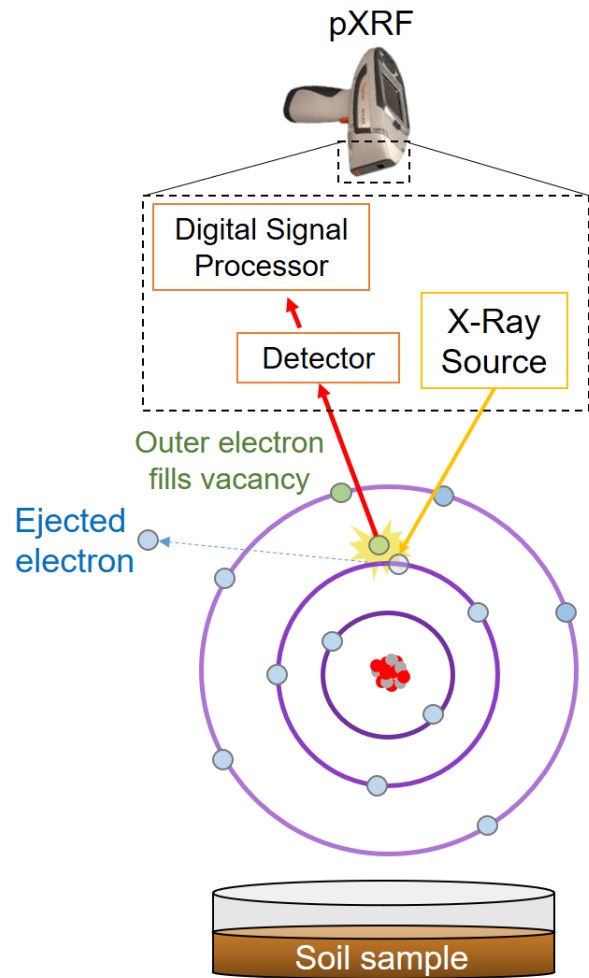



**Appendix B**

**Fig. B1**: Principles of the mean-bootstrapping technique to calculate mineral element stock in Yedoma domain deposits illustrated for Fe in Yedoma (Y). The bootstrapping statistical method use resampled (10 000 times) observed values (circled in red; i.e., mineral element concentration, bulk density, deposits thickness and ice-wedge volume (WIV)) and derive the mean afterward. This bootstrapping technique is used due to the non-normal distribution of the parameters. We used sampling with replacement, which means that after each step of the random draw from the original sample, we put the observation back before the following step. The process is done with 10 000 steps from
which a distribution is obtained. To estimate mineral elements stocks, we use the arithmetic mean and standard deviation assuming normal distribution (Strauss et al., 2013).

| Fe [mg.kg⁻¹] | Bulk density [g.cm⁻³] | | YThickness [m] | | YWIV = 1- Ice volume [%] | | Yedoma coverage [km²] | | Yedoma Stock [Gt] |
|---|---|---|---|---|---|---|---|---|---|
| 32500 | 1.16 | | 24.8 | | 0.64 | | | | 148.6 |
| 40568 | 0.88 | | 18.5 | | 0.5184 | | | | 140.6 |
| 12060 | 1.458 | | 26.75 | | 0.64 | | | | 158.6 |
| 30897 | 1.45 | X | 12.3 | X | 0.4096 | X | 410 000 | = | 149.6 |
| 48560 | 0.85 | | 8.4 | | 0.4096 | | | | 160.2 |
| 36158 | 0.994 | | 7.5 | | 0.5 | | | | 130.6 |
| .... | .... | | .... | | .... | | | | … |
| n = 814 | | | n = 19 | | n = 18 | | | | n = 10 000 |




**Appendix C**

**Fig. C1**: Lithology of the bedrock underlying (a) Yedoma deposits (n=814) and (b) Alas deposits (n=470) inferred from the Global Lithological Map (GLiM; Hartmann and Moosdorf, 2012).

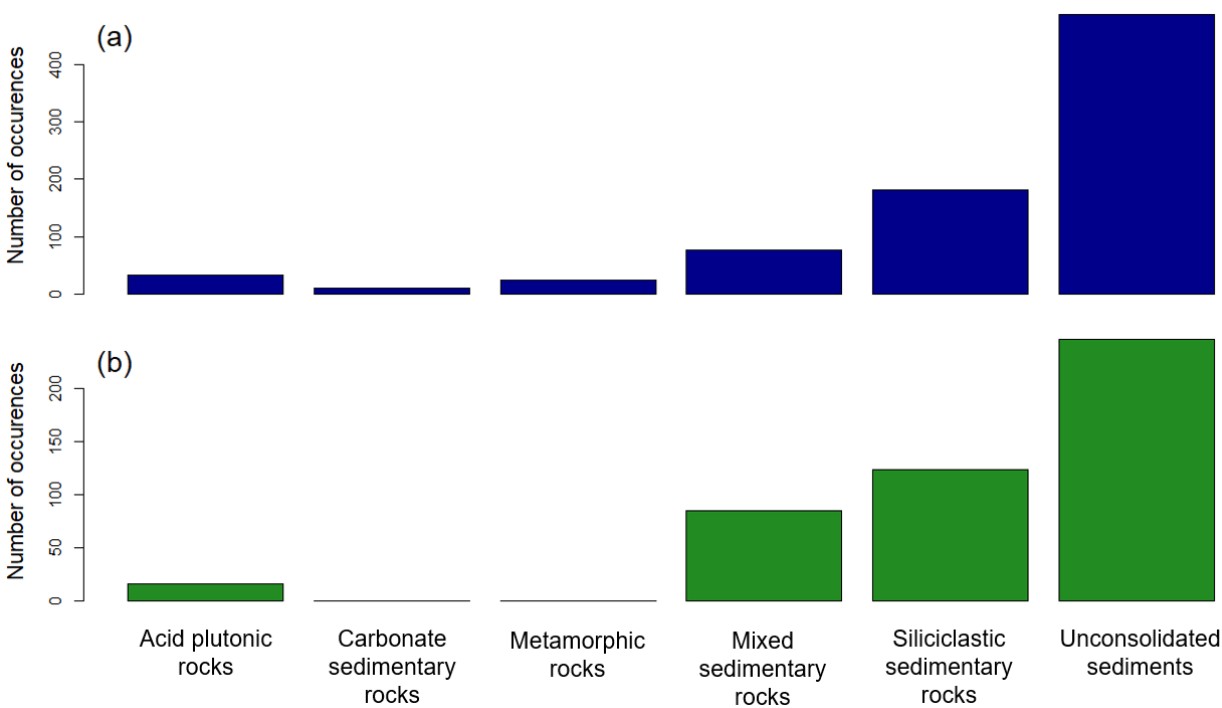




## Appendix D

**Table D1**: Mineral elements stock (kg.m$^{-3}$) summary in Yedoma and Alas deposits (n=814 and 470, respectively). This stock neglects the presence of ice wedges.

| | Stock (kg.m$^{-3}$) | | | |
| --- | --- | --- | --- | --- |
| | Yedoma deposits | | Alas deposits | |
| Element | Mean | ± 2σ | Mean | ± 2σ |
| Si | 345.6 | 8.34 | 335.3 | 14.07 |
| Al | 75.4 | 1.74 | 73.0 | 2.96 |
| Fe | 36.0 | 0.843 | 35.3 | 1.38 |
| Ca | 17.3 | 0.916 | 13.1 | 1.02 |
| K | 22.0 | 0.572 | 21.3 | 0.98 |
| Ti | 4.64 | 0.107 | 4.42 | 0.18 |
| Mn | 0.626 | 0.0883 | 0.587 | 0.035 |
| Zn | 0.080 | 0.0019 | 0.078 | 0.0039 |
| Sr | 0.235 | 0.0085 | 0.235 | 0.016 |
| Zr | 0.332 | 0.0094 | 0.315 | 0.017 |






## 9   Authors contribution

AM, SO, and JS conceived the project. JS, GG, LS, MF and PK retrieved samples from Alaska and Siberia from many different
field expeditions. AM did the pXRF measurements with the help of EM. AM, SO, EM, BP and AV contributed to set up the
YMCA database. AM analyzed the data and calculated to the stocks based on the code developed by JS for carbon stock
estimation using mean-bootstrapping. AM prepared the manuscript with contributions from all co-authors.

## 10   Competing interests

The authors declare that they have no conflict of interest.

## 11   Acknowledgments


The authors acknowledge Hélène Dailly and Anne Iserentant from the MOCA analytical platform at UCLouvain for mineral
elemental analysis. We thank Waldemar Schneider (AWI logistics) and Dmitry Melnichenko (Hydrobase Tiksi) for decadal
logistical support, and Catherine Hirst, Maxime Thomas, and Pierre Delmelle for fruitful scientific discussions. This project
received funding from the European Union's Horizon 2020 research and innovation program under grant agreement No.
714617 to SO (WeThaw). SO also acknowledges funding from the National Funds for Scientific Research FNRS (FC69480).
This work was embedded into the Action Group "The Yedoma Region" funded by the International Permafrost Association.
Recent expeditions were funded by a European Research Council (ERC) Starting Grant (338335) to GG as well as Alfred
Wegener Institutes basic funding. Kytalyk samples were collected with support of the European Union FP7—
ENVIRONMENT project PAGE21 (grant agreement 282700) and the EU FP7 INTERACT Integrating Activity.

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
