# Peer review of "Mineral element stocks in the Yedoma domain: a first assessment in ice-rich permafrost regions"

_Earth System Science Data, 2020_

## Referee Comment (RC1) · Anonymous Referee #1 · 21 Dec 2020

The manuscript is devoted to estimation of concentrations and stocks of some major and trace elements in mineral soils of the yedoma regions. The novelty of this work is not clear, the methodology of sampling, analysis and upscaling is not presented and there are a number of issues that have to be clarified before further evaluation of this manuscript becomes possible.

Major issues It is unclear, over which depth the stocks and concentrations are evaluated. The stocks are proportional to the volume so this should be clearly specified. Note that for soil C, the lateral pools are usually estimated as 0-30 cm, 0-100 cm and 0-300 cm. Here, similar distinction is needed.

The nature of mineral deposits is not discussed and unclear from the abstract - are these clays sands, shales?

[Figure]

The working hypothesis is missing. Why one would expect that sediments of yedoma be different from average world soils or sedimentary rocks (silts, shales)? Or from the world average alluvial or aeolian sediments? Might be so with some nutrients (N, P) but certainly not for conservative major and trace elements...A brief look at Table 4 demonstrates that the yedoma sediments are between world average soils (Bowen, 1979 Environmental chemistry of the elements) and shales/earth crust. Considering Q1-Q3 range of obtained values, the interest of conducting the whole work is unclear. In this regard, one could also estimate the storage of Ca, P, C in frozen sedimentary rocks of eastern Siberia, where the permafrost is 500 m deep. This permafrost may thaw in the future thus rising environmental concern.

The representativeness of samples is poor: only one site in Siberia and one site in Alaska are from the coast. In all samples, (past) marine sediments may influence the chemical composition

The use of BHVO-2 as certified material might not be suitable given some amount of OC. Instead/in addition, some LKSD or standard soil reference material is needed.

Section 3.2. Please provide the results of pXRF measurements using Niton X13tGoldd of BHVO-2, LKSD, and other soil certified materials whose mineral and organic composition roughly matches those of studied yedoma samples.

Table 2. The disagreement of Al, Fe, Ca, K and Sr strongly exceed any uncertainties. The reliability of pXRF measurements is in question.

The reasons of presenting correlations in Fig. 5 are unclear. Ca exhibits the highest correlation between two methods yet it is strongly (by a factor of 3.9) underestimated by pXRF (Table 2).

Specific comments L51-52 "associated to minerals" is unclear term. Does it imply stored in mineral (not peat) soils? Adsorbed onto /incorporated into mineral (such as Fe hydroxides or clays)?

L123-130 and Fig.3: The vertical scale is missing / unclear Table 1: Total deposits depth and sampled depth are missing.

L 153: One cannot use term 'soils' without providing soil sampling depth and soil classification. Were the samples subjected to < 2 mm sieving?

L234 elements in ice wedges. It is true that their concentration is negligible compared to that in quartz and clays. However, the lability (rate of release from reservoir to the hydrological network) of Ca, K, Fe from the ice is million (or billion?) times faster than that of refractory minerals. One cannot neglect ice wedges in this context of elementary transport from permafrost to rivers and soils within the thawing scenario

L258-259 please provide the depth of these deposits, otherwise the surface area estimation is irrelevant

Table 3: Slope of dependences should be provided

L290 This is relative uncertainty. Correctness of analysis is not discussed. Add slopes or 1:1 line to Fig. 5

L317-318: Table 3 does not allow to obtain reliable element concentrations; a linear equation with slope and intercept should be presented.

L375-377: Depth of sediments is absolutely needed, otherwise the vulnerability cannot be assessed.

L420-242: What about other highly labile elements such as Li, B, Mg, Na, K, Sr, Ba?

L428. Ca solubility is improper term (unless Ca is considered as native metal). One can speak about $CaCO_3$, $CaSO_4$ solubility. Otherwise use the term "mobility" or "lability" Section 5.3.1 is too general and irrelevant to the main findings of this study. Instead, based on mineral composition of studied yedoma soils, one can attempt to estimate the capacity of soil to capture organic matter. One can also explore element : C ratio to reveal which elements are most important for OC storage (unlikely that it would be

Si, Mn, Sr. . .).

L435: These are not elements that interact with OC but minerals. More specifically, surface sites of minerals ($>AlOH_2+$, $>FeOH_2+$, $>CaOH_2+$) can adsorb negatively charged organic ligands.

L459-460: The primary nutrients in this context are N and P

L480: depth is needed

L485-486: vertical distribution is not shown in the paper; why it is presented in the Conclusions?

L490-493 Compared to fully instantaneous mobility of elements from ice wedges, the amount of elements that can be released from minerals such quartz or clays might be negligible at the time scale of decades.

The term 'stock' is highly confusing: in Fig B1 it is in Gt, but in Table D1 it is in kg/m3. These are totally different units.

Fig. B1: L510+513 contradict to L515-516

―――――――――――――――

---

## Referee Comment (RC2) · Anonymous Referee #2 · 4 Jan 2021

I appreciate the efforts from the authors. I understand the authors created a valuable dataset for the mineral elements in the yedoma regions, and they also tried to calculated the storage of these elements. I have some comments for the authors to improve the quality of the manuscripts.

1. When the authors introduce the stocks or storage, it is necessary to clarify the depth or thickness of yedoma. At least, the authors should explain the characteristics of yedoma. This is important because the potential readers will be confused about the depth and height in the dataset.

2. It is difficult to follow in the sampling sites section. What is the more than 20 years of sampling? Does that mean the samples were collected during the past 20 years? The authors explained 22 locations (or areas), and total 1292 samples, but did not present

how many sites or profiles. How many soil profiles were measured? In the table 1, the authors introduced this, but is also necessary to explain the number of samples for each location and each yedoma profile. Just briefly introduce this.

3. I did not check the references listed in the Table 1. What did these references mean? These references were conducted in the location (or area), or include environmental conditions for these areas? I suggest the authors add some information about the landform in the table, and so the readers can understand why you select the number of profiles.

4. For the 3.1 and 4.1 sections, the authors claimed they analyzed 144 samples using ICP, what are the 144 samples? What locations, profiles, depths or heights?

5 Table 4, why the negative value for Si content?

6. In the discussion section, I would like to see the disadvantage of pXRF, so the future users can be careful use this dataset. I do not know if there were similar datasets from pXFR, but I know some soil scientists do not believe this method, the authors should also explain why this dataset are valuable for future studies.

7. I think it is necessary to compare the results with existing reports of the elements contents and stocks.

8. The authors calculated the stocks and storage of these elements based on the pXFR methods and the distribution data of yedema. The uncertainties in the yedoma distribution itself should be clarified.

9. I encourage the authors dig deeper about the relationship among the elements. For example, compare their contents and distributions with other soils, especially with soils in permafrost regions. These will be more interesting that the implications of the elements release in the present version. For what I see, it is a little speculative.

---

## Author Comment (AC1) · 1 Mar 2021

RC= Reviewer comment ; AR= Authors response

**RC**: I appreciate the efforts from the authors. I understand the authors created a valuable dataset for the mineral elements in the yedoma regions, and they also tried to calculated the storage of these elements. I have some comments for the authors to improve the quality of the manuscripts.

We thank the reviewer for the valuable comments and suggestions to improve the manuscript. We have revised the manuscript accordingly. Please find the details in the responses to the following comments.

**RC1**: When the authors introduce the stocks or storage, it is necessary to clarify the depth or thickness of yedoma. At least, the authors should explain the characteristics of yedoma. This is important because the potential readers will be confused about the depth and height in the dataset.

**AR** : We agree that the choice of the thickness used to upscale to the whole Yedoma domain was not clear in the manuscript. Here, mineral element stocks are compared with C stocks using identical Yedoma domain deposits parameters (including thicknesses) like in Strauss et al., 2013 for deep permafrost carbon pool of the Yedoma region, i.e., a mean thickness of 19.6 meters deep in Yedoma deposits and 5.5 meters deep in Alas deposits. We have revised the manuscript to include that information (L 282):" Thickness used for mineral element stock estimations in Yedoma domain deposits are based on mean profile depths of the sampled Yedoma (n=19) and Alas (n=10) deposits (Table 3; Strauss et al., 2013). Since the 10 000 step bootstrapping technique randomly picks one thickness at a time with replacement, we evaluated the stock estimation for a mean thickness of 19.6 meters deep in Yedoma deposits."

In addition, Table 3 has been added in the text to clarify the thickness used in the stock estimation.

|        | Yedoma        | deposits   | Alas deposits |            |  |  |
|--------|---------------|------------|---------------|------------|--|--|
|        | Thickness (m) | WIV (vol%) | Thickness (m) | WIV (vol%) |  |  |
| Mean   | 19.6          | 49.1       | 5.5           | 7.8        |  |  |
| Median | 15.1          | 51.9       | 4.6           | 7.6        |  |  |
| Min    | 4.6           | 34.7       | 1.2           | 0.8        |  |  |
| Max    | 46            | 59         | 13.4          | 12.8       |  |  |
| n      | 19            | 10         | 10            | 7          |  |  |

**Table 3**: Summary of key parameters for Yedoma deposits and Alas deposits. Parameters include thicknesses (in meters) and wedge ice volume (WIV, in %) from Strauss et al., 2013. Mineral element stock estimations are based on a mean thickness of 19.6 m for Yedoma and 5.5 m for Alas deposits.

**RC2**: It is difficult to follow in the sampling sites section. What is the more than 20 years of sampling? Does that mean the samples were collected during the past 20 years? The authors explained 22 locations (or areas), and total 1292 samples, but did not present how many sites or profiles. How many soil profiles were measured? In the table 1, the authors introduced this, but is also necessary to explain the number of samples for each location and each yedoma profile. Just briefly introduce this.

**AR**: We have revised the manuscript according to the comment from the reviewer. A table (Table A1; Appendix A) has been added to provide i) the number of profiles sampled for each location; ii) the number of samples analyzed with pXRF method and iii) the number of samples analyzed with ICP-OES method, and iv) the number of samples analyzed by X-ray diffraction as well as general geomorphology of each site. We have also revised the text (Section 2.2, L138-140): "More specifically, the dataset includes 22 locations and compiles 75 different profiles from West, North and Interior Alaska, the Kolyma region, the Indigirka region, the New Siberian Archipelago, the Laptev Sea coastal regions, and Central Yakutia (Fig. 1)."

**RC3**: I did not check the references listed in the Table 1. What did these references mean? These references were conducted in the location (or area), or include environmental conditions for these areas? I suggest the authors add some information about the landform in the table, and so the readers can understand why you select the number of profiles.

**AR**: Each reference in Table 1 provides detailed description of landforms, cryostratigraphy and paleoenvironmental characteristics of the sites and the sampled profiles. This has now been clarified in the caption of Table 1. We also added a more detailed Table (Appendix A) to present specific characteristics of the sites in each location.

**RC4**: For the 3.1 and 4.1 sections, the authors claimed they analyzed 144 samples using ICP, what are the 144 samples? What locations, profiles, depths or heights?

**AR**: We have revised the manuscript to add this information. In Table A1 (Appendix A), a specific column specifies how many samples were analyzed using ICP-OES measurements for each location. In Table C1 (Appendix C of the revised manuscript), we provide a list of the samples selected for linear regressions (n=144) with associated depth or height included.

RC5: Table 4, why the negative value for Si content?

**AR**: As specified in the caption of Table 4, the negative value is due to very high TOC content: "For Si, excessive organic rich samples may result in lower concentration values and consequently in negative concentration value after correction. Only positive values have been considered for stock calculations." We have now included a \* to refer to this information in the caption to clarify the point raised by the reviewer.

**RC6**: In the discussion section, I would like to see the disadvantage of pXRF, so the future users can be careful use this dataset. I do not know if there were similar datasets from pXFR, but I know some soil scientists do not believe this method, the authors should also explain why this dataset are valuable for future studies.

**AR**: A specific section discussing the advantages and the limitations of the pXRF method can be found in the revised version of the manuscript (section 5 in revised manuscript). The text emphasizes that *"The main limitation of the pXRF method is that raw concentration data cannot be used before applying a linear regression to correct for systematic error. This drawback related to inaccurate raw pXRF measurements can be rectified by correcting the raw pXRF values with accurate values obtained with the ICP-OES method after alkaline fusion (calibration based on 144 samples in this study)". As explained in the revised manuscript (line 220-224, and revised Figure 4), raw pXRF measurements have low trueness (i.e. high systematic error) compare to true value. This is why raw pXRF measurements must be corrected using a calibration obtained with a complementary accurate method. In our case, this correction is made through a calibration line for each element comparing XRF measurements and ICP-*

OES measurements after alkaline fusion on a subset of 144 samples (about 11% of the dataset). This correction is essential in order to use pXRF device is to obtain a reliable precision (i.e., low random errors; Table 2).

**Figure 4**: **(a)** Comparison between two methods to assess mineral element concentrations in deposit samples: the inductively coupled plasma optical-emission spectrometry (ICP-OES) method (large blue arrow) is destructive and time-consuming (involving several steps), whereas the portable X-ray fluorescence (pXRF) method (dotted red arrows) is non-destructive and allows a fast and reliable determination of element concentrations on a large set of samples in a cost-effective way when a correction with a linear regression is applied. **(b)** The pXRF bias (i.e., systematic error) is corrected with a linear regression specific to each pXRF device. Portable XRF devices from each lab require their own linear regression to correct for accurate value. The linear regression is obtained from a selection of samples (here 11 % of the samples) analysed by both methods, pXRF and ICP-OES.

In his/her comment, the reviewer asks for clarification about how this dataset is valuable for future studies. Section 6 has been revised in order to present the main directions in which the dataset can provide useful insights, i.e., to investigate the evolution of mineral-organic carbon interactions (new Figure 8), and the influence of mineral weathering on the carbon cycle and nutrient supply to ecosystems. It has also been included in the revised version (Line 534-536 in revised manuscript) that we acknowledge that *"The YMCA dataset is a first step needed in order to evaluate the impact of widespread rapid permafrost thaw through thermokarst processes (Turetsky et al., 2020) on the mineral element concentrations in the deposits and the potential implications for OC and mineral nutrient supply."*

**RC7**: I think it is necessary to compare the results with existing reports of the elements contents and stocks.

**AR**: To our knowledge, no mineral element stock assessment or mineral concentration assessment in Yedoma domain deposits exists. Some studies present the abundance of heavy mineral composition within some specific deposits profiles (Schirrmeister et al., 2002, 2003, 2008, 2010, 2011) but with no mineral concentration or stock assessment. In permafrost soils, the focus has been on soil properties such as exchangeable cations, cation exchange capacity (CEC), soil acidity, etc. (Ping et al., 1998, 2005). We have included references to existing work in permafrost soils in section 6.2 (Line 516).

**RC8**: The authors calculated the stocks and storage of these elements based on the pXFR methods and the distribution data of yedema. The uncertainties in the Yedoma distribution itself should be clarified.

**AR**: We agree that details were missing on how the total coverage of Yedoma and Alas deposits was estimated. We have revised the manuscript accordingly (L292-297 in revised manuscript):

"The core Yedoma domain extent was estimated to ~1 387 000 km2, based on digital Siberian Yedoma region map (Romanovskii, 1993) and the distribution of Alaskan ice-rich silt deposits equivalent to Yedoma (Jorgenson et al., 2008). The Yedoma deposit extent is estimated to 410 000 km2, i.e., 30% of the Yedoma domain, based on the fact that 70% of the Yedoma domain area is affected by degradation (Strauss et al., 2013). Considering 10% of the area of the Yedoma domain covered with lakes and rivers, 4% covered with other deposits including deltaic and fluvial unfrozen sediments, this leaves 56% (780 000 km2) of the Yedoma domain covered by frozen thermokarst deposits in drained thermokarst lakes (Fig. 2)."

**RC9**: I encourage the authors dig deeper about the relationship among the elements. For example, compare their contents and distributions with other soils, especially with soils in permafrost regions. These will be more interesting that the implications of the elements release in the present version. For what I see, it is a little speculative

**AR**: To address the comment from the reviewer, we have revised section 6. In section 6.1, we have included a new figure (Figure 8) to investigate the evolution of Fe to organic carbon ratio, and Al to organic carbon ratio upon thawing between Yedoma and Alas deposits.

In section 6.2, we consider the mineralogy of the deposits to investigate the evolution of mineral element concentration upon thawing between Yedoma and Alas deposits. This is the reason why we have included X-ray diffraction data providing the mineralogy of 40 samples from the Yedoma domain deposits (Table 6 here below). The associated methodology (section 3.3 in the revised manuscript) and results description (section 4.3 in the revised manuscript) have been included.

Table 6: Mineralogical composition in Yedoma (Y) and Alas and fluvial (A) deposits of Siberia and Alaska (Q, Quartz; PI, Plagioclase; Ch, Chlorite; M, Mica; I, Illite; K, Kaolinite; D, Dolomite; Ca, Calcite; Sm, Smectites). The diffractograms for each profile (n = number of diffractogram per profile) are presented in Appendix G (a-b-c-d-e-f-g, respectively).

| Site       | n  | Fraction | Profile label                               | Mineralogy          | App. G |
|------------|----|----------|---------------------------------------------|---------------------|--------|
| Sobo Sise  | 13 | Bulk     | Sobo T2-2 (Y), T2-3 (Y), T2-5 (A), T2-6 (A) | Q, Pl, Ch, M, K     | a)     |
| Buor Khaya | 3  | Bulk     | Buo-02 (Y)                                  | Q, Pl, Ch, M, K     | b)     |
| Buor Khaya | 6  | Bulk     | Buo-04 (Y)                                  | Q, Pl, Ch, M, K     | c)     |
| Buor Khaya | 1  | Clay     | Buo-04-A-01 (Y)                             | Q, Pl, Ch, I, K, Sm | d)     |
| Kytalyk    | 4  | Bulk     | KY T1-1 (Y), KY T2-2 (A)                    | Q, Pl, Ch, M, K     | e)     |
| Colville   | 3  | Bulk     | Col (Y)                                     | Q, PI, M, K, D, Ca  | f)     |
| Itkillik   | 10 | Bulk     | ltk (Y)                                     | Q, PI, M, K, D, Ca  | g)     |

Appendix A: Studied locations from the Yedoma domain with associated labels, total number of sampled profiles, number of samples analyzed with portable X-ray Fluorescence (pXRF), inductively coupled plasma optical emission spectrometry (ICP-OES) after alkaline fusion and X-ray diffraction method (XRD). A simplified geomorphological description of each sampling location is presented (detailed information is provided in the reference papers cited in Table 1). The site numbers (Site Nb) 1-17 are from Siberia, and 18-22 are from Alaska.

| Site Nb | Site Name                     | Label       | Total sampled profiles | Samples analyzed with
pXRF method | Samples analyzed with
ICP-OES method | Samples characterized with
XRD method | Geomorphology/
Landform                                                                                                                            |
|---------|-------------------------------|-------------|------------------------|--------------------------------------|-----------------------------------------|------------------------------------------|-------------------------------------------------------------------------------------------------------------------------------------------------------|
| 1       | Cape Mamontov Klyk            | Mak         | 3                      | 80                                   | -                                       | -                                        | Sedimentary coastal plain                                                                                                                             |
| 2       | Nagym (Ebe Sise Island)       | Nag         | 3                      | 29                                   | -                                       | -                                        | Cliff section with thermokarst mounds                                                                                                                 |
| 3       | Khardang Island               | Kha         | 1                      | 31                                   | 1                                       | -                                        | Cliff section with thermokarst mounds                                                                                                                 |
| 4       | Kurungnakh Island             | Bkh,
KUR | 4                      | 143                                  | 2                                       | -                                        | Fragment of a broad foreland plain north of the Chekanovsky Ridge                                                                                     |
| 5       | Sobo Sise Island              | Sob         | 4                      | 58                                   | 58                                      | 13                                       | Yedoma uplands but also features permafrost
degradation landforms (thermokarst lakes,
drained thaw lake basin, and thermo-erosional gullies)    |
| 6       | Bykovsky Peninsula            | Mkh,
BYK | 7                      | 150                                  | 2                                       | -                                        | Remnants of an accumulation plain                                                                                                                     |
| 7       | Muostakh Island               | Muo         | 1                      | 11                                   | -                                       | -                                        | Remnants of an accumulation plain                                                                                                                     |
| 8       | Buor Khaya Peninsula          | Buo         | 5                      | 80                                   | 44                                      | 10                                       | Coastal lowlands - late Pleistocene accumulation plains                                                                                               |
| 9       | Stolbovoy Island              | Sto         | 4                      | 16                                   | 1                                       | -                                        | Step-like cryoplanation terraces with several levels                                                                                                  |
| 10      | Belkovsky Island              | Bel         | 2                      | 12                                   | -                                       | -                                        | Step-like cryoplanation terraces with several levels                                                                                                  |
| 11      | Kotel'ny Island               | KyS         | 1                      | 10                                   | -                                       | -                                        | Step-like cryoplanation terraces with several levels                                                                                                  |
| 12      | Bunge Land                    | Bun         | 1                      | 8                                    | -                                       | -                                        | More-or-less homogeneous flat sandy plain                                                                                                             |
| 13      | Bol'shoy Lyakhovsky
Island | TZ, R, L    | 15                     | 150                                  | 3                                       | -                                        | Gradually sloping terrain intersected by rivers and thermo-erosional valleys                                                                          |
| 14      | Oyogos Yar coast              | Oy          | 2                      | 50                                   | 1                                       | -                                        | Very gently inclined step-like surface of the Yana-Indigirka Lowland                                                                                  |
| 15      | Kytalyk                       | KY, KH      | 3                      | 50                                   | 4                                       | 4                                        | Recent and sub-recent floodplains, Yedoma and Alas                                                                                                    |
| 16      | Duvanny Yar                   | DY          | 6                      | 143                                  | 5                                       | -                                        | Hills dissected by deep thermos-erosional valleys and thermokarst depressions                                                                         |
| 17      | Yukechi                       | Yuk-Yul     | 4                      | 87                                   | 2                                       | -                                        | Yedoma uplands and drained alas basins, indicating active thermokarst processes                                                                       |
| 18      | Kitluk                        | Kit         | 2                      | 45                                   | 2                                       | -                                        | Tundra-covered coastal plain                                                                                                                          |
| 19      | Baldwin Peninsula             | Bal         | 4                      | 70                                   | 1                                       | -                                        | Sequence of marine, fluvial and glaciogenic sediments, which are well exposed along coastal bluffs and in some regions covered by loess-like deposits |
| 20      | Colville                      | Col         | 1                      | 23                                   | 8                                       | 3                                        | High exposure along the Colville River                                                                                                                |
| 21      | Itkillik                      | Itk, It     | 1                      | 22                                   | 10                                      | 10                                       | High exposure along the lower Itkillik River formed by active river erosion of a large remnant of originally gently undulating yedoma terrain         |
| 22      | Vault Creek Tunnel            | FAI         | 1                      | 24                                   | -                                       | -                                        | Permafrost tunnel about 40 m deep and 220 m long on north facing slope                                                                                |
| TOTAL   | 22                            |             | 75                     | 1292                                 | 144                                     | 40                                       |                                                                                                                                                       |

**Appendix C**: List of samples (n=144) analysed by both inductively coupled plasma optical-emission spectrometry (ICP-OES) and portable X-ray fluorescence (XRF) method for linear regressions determination. Depth (below surface level) or \*height (above sea level) are provided (in meters) if available. Labels are associated to labels from the Yedoma domain Mineral Concentration Assessment (YMCA) dataset for additional characteristics

| n  | Sample label  | Depth/  | n  | Sample label  | Depth/  | n   | Sample label     | Depth/  |
|----|---------------|---------|----|---------------|---------|-----|------------------|---------|
|    |               | Height* |    |               | Height* |     |                  | Height* |
|    |               | (m)     |    |               | (m)     |     |                  | (m)     |
| 1  | Sob14-T2-2-03 | 0.225   | 49 | Sob14-T2-6-09 | 0.775   | 97  | Buo-02-D-23      | 7       |
| 2  | Sob14-T2-2-05 | 0.43    | 50 | Sob14-T2-6-11 | 0.96    | 98  | Col-5-1          | 1.8     |
| 3  | Sob14-T2-2-07 | 0.6     | 51 | Sob14-T2-6-13 | 1.085   | 99  | Col-5-6          | 3.02    |
| 4  | Sob14-T2-2-09 | 0.745   | 52 | Sob14-T2-6-15 | 1.15    | 100 | Col-5-10         | 5.4     |
| 5  | Sob14-T2-2-11 | 0.87    | 53 | Sob14-T2-6-16 | 1.2     | 101 | Col-5-13         | 6.9     |
| 6  | Sob14-T2-2-13 | 1.015   | 54 | Buo-04-A-00   | 0.1     | 102 | Col-5-15         | 9       |
| 7  | Sob14-T2-2-15 | 1.15    | 55 | Buo-04-A-01   | 1       | 103 | Col-5-18         | 11.6    |
| 8  | Sob14-T2-2-17 | 1.25    | 56 | Buo-04-A-02   | 1.5     | 104 | lt-1             | 2.3     |
| 9  | Sob14-T2-2-19 | 1.4     | 57 | Buo-04-A-03   | 2       | 105 | lt-6             | 5.4     |
| 10 | Sob14-T2-2-21 | 1.565   | 58 | Buo-04-A-04   | 2.5     | 106 | ltk-E-03         | 9.2     |
| 11 | Sob14-T2-2-23 | 1.7     | 59 | Buo-04-A-05   | 3       | 107 | ltk-H-02         | 13.7    |
| 12 | Sob14-T2-2-25 | 1.82    | 60 | Buo-04-A-06   | 3.5     | 108 | ltk-F-03         | 19.5    |
| 13 | Sob14-T2-2-27 | 1.935   | 61 | Buo-04-A-07   | 4.5     | 109 | 14C-1            | 20.6    |
| 14 | Sob14-T2-2-29 | 2.14    | 62 | Buo-04-A-08   | 5       | 110 | ltk-D-06         | 23.8    |
| 15 | Sob14-T2-2-31 | 2.3     | 63 | Buo-04-B-09   | 8       | 111 | ltk-C-02         | 25.9    |
| 16 | Sob14-T2-3-03 | 0.265   | 64 | Buo-04-B-10   | 8.5     | 112 | ltk-B-02         | 27.5    |
| 17 | Sob14-T2-3-05 | 0.52    | 65 | Buo-04-B-11   | 9       | 113 | ltk-J-02         | 28.9    |
| 18 | Sob14-T2-3-07 | 0.66    | 66 | Buo-04-B-12   | 9.5     | 114 | Sob14-T2-2-03bis | 0.225   |
| 19 | Sob14-T2-3-09 | 0.835   | 67 | Buo-04-B-13   | 10      | 115 | Sob14-T2-2-07bis | 0.6     |
| 20 | Sob14-T2-3-11 | 1.045   | 68 | Buo-04-C-14   | 9.5     | 116 | Sob14-T2-2-15bis | 1.15    |
| 21 | Sob14-T2-3-13 | 1.22    | 69 | Buo-04-C-15   | 10      | 117 | Sob14-T2-2-20bis | 1.5     |
| 22 | Sob14-T2-3-15 | 1.45    | 70 | Buo-04-C-16   | 10.5    | 118 | Sob14-T2-2-30bis | 2.2     |
| 23 | Sob14-T2-3-17 | 1.685   | 71 | Buo-04-C-17   | 11      | 119 | KY-T1-1-9-14     | 0.125   |
| 24 | Sob14-T2-3-19 | 1.885   | 72 | Buo-04-C-19   | 11.7    | 120 | KY-T1-1-90-95    | 0.925   |
| 25 | Sob14-T2-3-21 | 2.04    | 73 | Buo-04-C-20   | 12      | 121 | KY-T2-2-27-32    | 0.295   |
| 26 | Sob14-T2-3-23 | 2.2     | 74 | Buo-04-C-21   | 12.5    | 122 | KY-T2-2-94-100   | 0.97    |
| 27 | Sob14-T2-3-25 | 2.4     | 75 | Buo-04-C-22   | 13      | 123 | Oy7-11-16        | 11.9*   |
| 28 | Sob14-T2-5-03 | 0.125   | 76 | Buo-04-C-23   | 13.5    | 124 | 52Mkh-KB-7-5     | 22.3*   |
| 29 | Sob14-T2-5-05 | 0.375   | 77 | Buo-02-A-01   | 0.3     | 125 | DY-01-F-34       | 29.1*   |
| 30 | Sob14-T2-5-07 | 0.48    | 78 | Buo-02-A-02   | 0.6     | 126 | DY-02-A-01       | 5*      |
| 31 | Sob14-T2-5-09 | 0.635   | 79 | Buo-02-A-03   | 0.7     | 127 | DY-04-A-01       | 7.85*   |
| 32 | Sob14-T2-5-11 | 0.775   | 80 | Buo-02-A-04   | 1.2     | 128 | DY-04-A-02       | 7.7*    |
| 33 | Sob14-T2-5-13 | 0.945   | 81 | Buo-02-A-05   | 1.7     | 129 | DY-05-B-05       | 2.7*    |
| 34 | Sob14-T2-5-15 | 1.15    | 82 | Buo-02-A-06   | 2.2     | 130 | Kit-8-5          | -       |
| 35 | Sob14-T2-5-17 | 1.32    | 83 | Buo-02-B-07   | 2.5     | 131 | Col-5-2          | -       |
| 36 | Sob14-T2-5-19 | 1.505   | 84 | Buo-02-B-08   | 3       | 132 | Col-5-17         | -       |
| 37 | Sob14-T2-5-21 | 1.75    | 85 | Buo-02-B-09   | 3.5     | 133 | Sto-1-1          | 0.25    |
| 38 | Sob14-T2-5-23 | 2.03    | 86 | Buo-02-B-10   | 4       | 134 | 1TZ-2-2          | 17.65*  |
| 39 | Sob14-T2-5-25 | 2.19    | 87 | Buo-02-B-11   | 4.5     | 135 | L21+50-S-3       | 4.3*    |
| 40 | Sob14-T2-5-27 | 2.39    | 88 | Buo-02-B-13   | 5.5     | 136 | L7-08-03         | 4.5*    |
| 41 | Sob14-T2-5-29 | 2.57    | 89 | Buo-02-C-14   | 5.2     | 137 | 126Mkh-6.1.1     | 1.25*   |
| 42 | Sob14-T2-5-31 | 2.76    | 90 | Buo-02-C-15   | 5.7     | 138 | 11KH-3007-1-4    | 0.6     |
| 43 | Sob14-T2-5-33 | 2.94    | 91 | Buo-02-C-16   | 6.3     | 139 | Bkh2002 S17      | 32.5*   |
| 44 | Sob14-T2-5-35 | 3.075   | 92 | Buo-02-C-17   | 6.9     | 140 | BAL16-B2-30      | 10.97   |
| 45 | Sob14-T2-5-36 | 3.1525  | 93 | Buo-02-D-18   | 4.5     | 141 | YUK15-YUL7-5     | 7.75*   |
| 46 | Sob14-T2-6-03 | 0.125   | 94 | Buo-02-D-19   | 5       | 142 | YUK15-YUL7-15    | 17.96*  |
| 47 | Sob14-T2-6-05 | 0.325   | 95 | Buo-02-D-20   | 5.5     | 143 | K-10-14-4        | 14.38   |
| 48 | Sob14-T2-6-07 | 0.525   | 96 | Buo-02-D-22   | 6.5     | 144 | Kit-7-2          | -       |

Jorgenson, M. T., Yoshikawa, K., Kanveskiy, M., Shur, Y., Romanovsky, V., Marchenko, S., Grosse, G., Brown, J. and Jones, B.: Permafrost characteristics of Alaska, 2008.

Ping, C. L., Bockheim, J. G., Kimble, J. M., Michaelson, G. J. and Walker, D. A.: Characteristics of cryogenic soils along a latitudinal transect in arctic Alaska, J. Geophys. Res. Atmospheres, 103(D22), 28917–28928, https://doi.org/10.1029/98JD02024, 1998.

Ping, C.-L., Michaelson, G. J., Kimble, J. M. and Walker, D. A.: Soil Acidity and Exchange Properties of Cryogenic Soils in Arctic Alaska, Soil Sci. Plant Nutr., 51(5), 649–653, https://doi.org/10.1111/j.1747-0765.2005.tb00083.x, 2005.

Romanovskii, N. N.: Fundamentals of cryogenesis of lithosphere, Moscow University Press, Moscow., 1993.

Schirrmeister, L., Siegert, C., Kunitzky, V. V., Grootes, P. M. and Erlenkeuser, H.: Late Quaternary ice-rich permafrost sequences as a paleoenvironmental archive for the Laptev Sea Region in northern Siberia, Int. J. Earth Sci., 91(1), 154–167, https://doi.org/10.1007/s005310100205, 2002.

Schirrmeister, L., Grosse, G., Kunitsky, V. V., Fuchs, M. C., Krbetschek, M., Andreev, A. A., Herzschuh, U., Babyi, O., Siegert, C., Meyer, H., Derevyagin, A. Y. and Wetterich, S.: The mystery of Bunge Land (New Siberian Archipelago): implications for its formation based on palaeoenvironmental records, geomorphology, and remote sensing, Quat. Sci. Rev., 29(25–26), 3598–3614, https://doi.org/10.1016/j.quascirev.2009.11.017, 2010.

Strauss, J., Schirrmeister, L., Grosse, G., Wetterich, S., Ulrich, M., Herzschuh, U. and Hubberten, H.-W.: The deep permafrost carbon pool of the Yedoma region in Siberia and Alaska: Deep carbon of Siberia and Alaska, Geophys. Res. Lett., 40(23), 6165–6170, https://doi.org/10.1002/2013GL058088, 2013.

Turetsky, M. R., Abbott, B. W., Jones, M. C., Anthony, K. W., Olefeldt, D., Schuur, E. A. G., Grosse, G., Kuhry, P., Hugelius, G., Koven, C., Lawrence, D. M., Gibson, C., Sannel, A. B. K. and McGuire, A. D.: Carbon release through abrupt permafrost thaw, Nat. Geosci., 13(2), 138–143, https://doi.org/10.1038/s41561-019-0526-0, 2020.

---

## Author Comment (AC2) · 1 Mar 2021

**To Anonymous Referee #1**

**RC**= Reviewer comment ; **AR**= Authors response

**RC:** The manuscript is devoted to estimation of concentrations and stocks of some major and trace elements in mineral soils of the yedoma regions. The novelty of this work is not clear, the methodology of sampling, analysis and upscaling is not presented and there are a number of issues that have to be clarified before further evaluation of this manuscript becomes possible.

**AR:** We thank the reviewer for the valuable detailed comments. We have carefully revised the manuscript to clarify the points that were unclear. Please find the details in the responses to the following comments. Concerning the raised point of novelty, we want to highlight that there is no comparable data for the permafrost region published.

**Major issues**

**RC1:** It is unclear, over which depth the stocks and concentrations are evaluated. The stocks are proportional to the volume so this should be clearly specified. Note that for soil C, the lateral pools are usually estimated as 0-30 cm, 0-100 cm and 0-300 cm. Here, similar distinction is needed.

**AR:** Thank you for this comment. With our mineral element stock estimate we used the depth and area published by Strauss et al., 2013. This is a mean thickness of 20 meters for Yedoma deposits and 5.5 meters deep for Alas deposits. This study and sampling over the last 25 years was not focused on the top meters of soil and therefore the separation into the suggested lateral pool is not possible.

Following the reviewers comment we have revised the manuscript to include that information (L 282-286):" *Thickness used for mineral element stock estimations in Yedoma domain deposits are based on mean profile depths of the sampled Yedoma (n=19) and Alas (n=10) deposits (Table 3; Strauss et al., 2013). Since the 10 000 step bootstrapping technique randomly picks one thickness at a time with replacement, we evaluated the stock estimation for a mean thickness of 19.6 meters deep in Yedoma deposits and 5.5 meters deep in Alas deposits.*"

In addition, Table 3 has been added in the text to clarify the thickness used in the stock estimation.

*Table 3: Summary of key parameters for Yedoma deposits and Alas deposits. Parameters include thicknesses (in meters) and wedge ice volume (WIV, in %) from Strauss et al., 2013. Mineral element stock estimations are based on a mean thickness of 19.6 m for Yedoma and 5.5 m for Alas deposits.*

|  | Yedoma deposits | | Alas deposits | |
|  | Thickness (m) | WIV (vol%) | Thickness (m) | WIV (vol%) |
| --- | --- | --- | --- | --- |
| Mean | 19.6 | 49.1 | 5.5 | 7.8 |
| Median | 15.1 | 51.9 | 4.6 | 7.6 |
| Min | 4.6 | 34.7 | 1.2 | 0.8 |
| Max | 46 | 59 | 13.4 | 12.8 |
| n | 19 | 10 | 10 | 7 |

**RC2:** The nature of mineral deposits is not discussed and unclear from the abstract – are these clays sands, shales?

**AR:** The comment of the reviewer is related to grain size (clay and sand fraction), lithology (shale) and mineralogy (nature of mineral deposits).

To clarify the grain-size of the deposit, we added (L 103-106) that : *"Yedoma deposits were first described as homogeneous silty fine, ice-, and organic-rich sediments derived from aeolian processes (i.e. loess or loess-related deposits) (Murton et al., 2015; Pewe and Journaux, 1983; Tomirdiaro and Chernen'kiy, 1987). Current evidence indicates that depositional processes are polygenetic including alluvial and aeolian deposition and re-deposition, as well as in situ weathering during the late Pleistocene cold stages (Schirrmeister et al., 2013). "*

The lithology of the bedrock underlying Yedoma and Alas deposits is presented in Appendix I (revised manuscript).

To clarify the mineralogy of the deposit, we have included X-ray diffraction data providing the mineralogy of 40 samples from five different locations from the Yedoma domain deposits (Table 6, here below; and Appendix G with the diffractograms). The associated methodology (section 3.3, here below) and results description (section 4.3, here below) have been included in the revised version.

In method section (3.3):

*"The X-ray diffraction (XRD) method allows the characterization of the presence of crystalline mineral phases. We assessed the mineralogy of 39 finely ground bulk samples and one clay fraction sample (samples selected in Siberia and Alaska detailed in Appendix A). The mineralogy of bulk samples was determined on powder (Cu Kα, Bruker Advance D8). Clay fraction mineralogy was assessed after $K^+$ and $Mg^{2+}$ saturation, ethylene glycol (eg) solvation and thermal treatments at 300 and 550°C (Robert and Tessier, 1974). The clay size fraction (<2 μm) was recovered after sonication, sieving at 50 μm to remove the sand fraction, and dispersion with $Na^+$-resins to separate silt and clay fractions (Rouiller et al., 1972)".*

In result section (4.3):

*"The main mineral phases (i.e., primary and secondary minerals) identified in selected Yedoma, Alas and fluvial deposits are presented in Table 6. The following minerals were identified in all bulk samples: quartz, feldspar plagioclase, micas and kaolinite. Chlorite was identified in Siberian deposits from Sobo Sise, Buor Khaya and Kytalyk. Calcite and dolomite were only detected in Alaskan Yedoma deposits from Colville and Itkillik. The mineralogy is generally similar along the profile depth for each location (Appendix G Fig. G1a-b-d-e-f-g). The diffractograms on the clay fraction from the Buor Khaya sample highlighted the presence of kaolinite, illite, and smectite (Appendix G Fig. G1c). These data highlight the co-existence in these deposits of highly stable minerals such as quartz with more weatherable minerals such as dolomite, calcite, or feldspar plagioclase (Cornelis et al., 2011; Goldich, 1938; Wilson, 2004), and the presence of clay minerals characterized by higher specific surface area relative to the other mineral constituents (i.e., smectite, kaolinite, illite) (Kahle et al., 2004; Saidy et al., 2012)".*

*Table 6: Mineralogical composition in Yedoma (Y) and Alas and fluvial (A) deposits of Siberia and Alaska (Q, Quartz; Pl, Plagioclase; Ch, Chlorite; M, Mica; I, Illite; K, Kaolinite; D, Dolomite; Ca, Calcite; Sm, Smectites). The diffractograms for each profile (n = number of diffractogram per profile) are presented in Appendix G (a-b-c-d-e-f-g, respectively).*

| Site | n | Fraction | Profile label | Mineralogy | Appendix G |
|------|---|----------|---------------|------------|------------|
| Sobo Sise | 13 | Bulk | Sobo T2-2 (Y), T2-3 (Y), T2-5 (A), T2-6 (A) | Q, Pl, Ch, M, K | a) |
| Buor Khaya | 3 | Bulk | Buo-02 (Y) | Q, Pl, Ch, M, K | b) |
| Buor Khaya | 6 | Bulk | Buo-04 (Y) | Q, Pl, Ch, M, K | c) |
| Buor Khaya | 1 | Clay | Buo-04-A-01 (Y) | Q, Pl, Ch, I, K, Sm | d) |
| Kytalyk | 4 | Bulk | KY T1-1 (Y), KY T2-2 (A) | Q, Pl, Ch, M, K | e) |
| Colville | 3 | Bulk | Col (Y) | Q, Pl, M, K, D, Ca | f) |
| Itkillik | 10 | Bulk | Itk (Y) | Q, Pl, M, K, D, Ca | g) |

**RC3:** The working hypothesis is missing. Why one would expect that sediments of Yedoma be different from average world soils or sedimentary rocks (silts, shales)? Or from the world average alluvial or aeolian sediments? Might be so with some nutrients (N, P) but certainly not for conservative major and trace elements. A brief look at Table 4 demonstrates that the yedoma sediments are between world average soils (Bowen, 1979 Environmental chemistry of the elements) and shales/earth crust. Considering Q1-Q3 range of obtained values, the interest of conducting the whole work is unclear. In this regard, one could also estimate the storage of Ca, P, C in frozen sedimentary rocks of eastern Siberia, where the permafrost is 500 m deep. This permafrost may thaw in the future thus rising environmental concern.

**AR:** We understand that the working hypothesis was unclear and this has now been clarified (L 74-78 in revised manuscript): "*Assessing the evolution of the mineral element stocks between never thawed (since deposition) and previously thawed, refrozen as well as newly formed deposits will contribute to a better understanding of the impact of past thawing processes on the evolution of the pool of mineral elements available for mineral-OC interactions. It also provides insights into how ongoing permafrost thaw and thermokarst processes may impact mineral elements and what the potential implications are for the fate of OC in ice-rich deposits (Strauss et al., 2017).*"

It has been included in the revised manuscript that this assessment is "*a first step needed in order to evaluate the impact of widespread rapid permafrost thaw through thermokarst processes (Turetsky et al., 2020) on the mineral element concentrations in the deposits and the potential implications for OC and mineral nutrient supply*" (L 534-536 in revised manuscript).

It is expected that Yedoma deposits would reflect the chemical composition of the Earth's crust given the contribution from glacial flour resulting from mixed lithologies (see map GUM, L 366).

Beyond the assessment of the global mineral element stocks in the Yedoma domain, it has been clarified in the revised version (section 6) that the dataset generated is used to investigate, for a given location, the evolution of the mineral element concentration with thermokarst processes, i.e., between Yedoma and Alas deposits (section 6, Figure 8 and Figure 9).

**RC4:** The representativeness of samples is poor: only one site in Siberia and one site in Alaska are from the coast. In all samples, (past) marine sediments may influence the chemical composition.

**AR:** Thank you, this is an important point. Taking today´s coastline, there are much more than two sites included from the coast (i.e., about eleven sites). Nevertheless, during Yedoma deposition a huge part of the Siberian Shelf was exposed, so the coastline was located several of hundreds kilometers to the north. Yedoma deposits do not include marine sediments (Schirrmeister et al 2013, Strauss et al 2017). Following the reviewers comment we revised the manuscript to include a table presenting the number of profiles for each location, a simplified geomorphological description of each site and number of sample analyzed with ICP-OES, pXRF and X-ray diffraction (Appendix A). The representativeness is based on a sampling scheme including two main Arctic regions (Alaska and Siberia) with as many lithologies (Appendix I), landforms (Appendix A), thawing history (Yedoma vs Alas) as possible. Several profiles have been collected in each "site", i.e., in each location (Table 1). The profile sampling was performed in order to cover as many geomorphologies, permafrost features (Yedoma, Alas deposits) within this particular site, detailed in reference papers (Table 1) and summarized in Appendix A (Table A1). Since there are no marine deposits in the Yedoma and Alas sequences, they cannot influence the chemical composition of the deposits. Only very young lagoon formations and modern seawater could have influence. However, lagoon sites were not considered here and against the seawater influence, the sediments were cleaned from thawed material during sampling. We agree with the reviewer that this study is a first order estimate and that, in the future, additional sampling points will be needed to create a more comprehensive dataset. This has been included in the revised version (L 534-536 in revised manuscript): "*The YMCA dataset is a first step needed in order to evaluate the impact of widespread rapid permafrost thaw through thermokarst processes (Turetsky et al., 2020) on the mineral element concentrations in the deposits and the potential implications for OC and mineral nutrient supply.*"

**RC5:** The use of BHVO-2 as certified material might not be suitable given some amount of OC. Instead/in addition, some LKSD or standard soil reference material is needed.

We have included measurements on soil reference materials (Table B1; Appendix B).

Table B1 : Accuracy on the measurement by inductively coupled plasma optical-emission spectrometry (ICP-OES) after alkaline fusion for three certified reference materials: i) USGS BHVO-2 (Wilson, 1997, p.199), ii) GBW07401 (GSS-1) and iii) GBW07404 (GSS-4) (National Research Centre for CRM, 1986). The mean and standard deviation (SD; n= number of repetitions) of ICP-OES values and certified values are shown. The offset, defined as the difference between certified and ICP-OES value over the certified value, expressed in %, is provided for each reference material.

| | Si | Al | Fe | Ca | K | Ti | Mn | Zn | Sr | Zr |
|---|---|---|---|---|---|---|---|---|---|---|
| Units | Wt % | Wt % | Wt % | Wt % | Wt % | Wt % | mg kg$^{-1}$ | mg kg$^{-1}$ | mg kg$^{-1}$ | mg kg$^{-1}$ |
| **BHVO-2** | | | | | | | | | | |
| n | 12 | 12 | 12 | 12 | 12 | 12 | 12 | 5 | 11 | 12 |
| ICP-OES mean | 23.2 | 7.16 | 8.56 | 8.06 | 0.43 | 1.61 | 1291 | 110 | 392 | 176 |
| (SD) | (0.2) | (0.07) | (0.14) | (0.11) | (0.03) | (0.02) | (49) | (7) | (29) | (9) |
| Certified values mean | 23.3 | 7.16 | 8.63 | 8.17 | 0.43 | 1.63 | 1290 | 103 | 389 | 172 |
| (SD) | (0.3) | (0.08) | (0.14) | (0.12) | (0.01) | (0.02) | (40) | (6) | (23) | (11) |
| Offset (%) | -0.3 | 0.02 | -0.8 | -1.4 | 0.5 | -1.2 | 0.05 | 7.2 | 0.8 | 2.4 |
| **GBW07401 (GSS-1)** | | | | | | | | | | |
| n | 5 | 5 | 5 | 5 | 5 | 5 | 5 | 5 | 5 | 5 |
| ICP-OES mean | 28.7 | 7.43 | 3.48 | 1.15 | 2.09 | 0.47 | 1703 | 666 | 152 | 239 |
| (SD) | (0.79) | (0.16) | (0.05) | (0.01) | (0.04) | (0.01) | (30) | (9) | (3) | (10) |

| | | | | | | | | | | |
|---|---|---|---|---|---|---|---|---|---|---|
| Certified values mean | 29.3 | 7.50 | 3.63 | 1.23 | 2.15 | 0.48 | 1760 | 680 | 155 | 245 |
| (SD) | (0.07) | (0.07) | (0.06) | (0.04) | (0.03) | (0.02) | (63) | (25) | (7) | (12) |
| Offset (%) | -2.0 | -0.9 | -4.2 | -6.2 | -2.6 | -2.8 | -3.2 | -2.1 | -1.9 | -2.3 |
| **GBW07404 (GSS-4)** | | | | | | | | | | |
| n | 5 | 5 | 5 | 5 | 5 | 5 | 5 | 5 | 5 | 5 |
| ICP-OES mean | 23.7 | 12.3 | 6.76 | 0.15 | 0.87 | 1.06 | 1404 | 209 | 78 | 518 |
| (SD) | (0.4) | (0.3) | (0.15) | (0.01) | (0.02) | (0.02) | (19) | (3) | (1) | (4) |
| Certified values mean | 23.8 | 12.4 | 7.20 | 0.19 | 0.86 | 1.08 | 1420 | 210 | 77 | 500 |
| (SD) | (0.1) | (0.1) | (0.08) | (0.03) | (0.05) | (0.03) | (75) | (13) | (6) | (42) |
| Offset (%) | -0.3 | -0.6 | -6.1 | -21 | 1.3 | -2.1 | -1.2 | -0.4 | 1.0 | 3.7 |

**RC6:** Section 3.2. Please provide the results of pXRF measurements using Niton X13tGoldd of BHVO-2, LKSD, and other soil certified materials whose mineral and organic composition roughly matches those of studied yedoma samples.

**AR:** This comment and the two following (and some specific comments) are related to the pXRF methodology for which we provide clarifications. We do not use raw pXRF measurements. Raw pXRF measurement must be corrected for trueness with an established calibration. This has been clarified in the revised version (L 220-221 in revised manuscript):

*"Raw pXRF concentrations cannot be used for absolute quantification if not corrected with a reliable and accurate method. Here, only pXRF concentration values corrected using a well-defined regression were used (Sect. 4.1)."*

With all methods, every measurement has a random and a systematic error. The accuracy of a measurement is characterized by its trueness (i.e., systematic error) and precision (i.e., random errors). A measurement with low random and low systematic error is therefore considered as accurate. Here, raw pXRF concentrations are assessed for their precision, regardless of the systematic error (i.e. bias). The systematic error made with pXRF method is corrected using a linear regression based on an accurate method (ICP-OES measurement after alkaline fusion). This linear regression is specific for each pXRF device, meaning that a pXRF device from another lab would need another linear regression to correct for trueness. This has been clarified in the revised manuscript with a new Figure 4 (L 225) and Line 449-451 in revised manuscript.

Certified materials are used in order to assess the trueness of a method, i.e., to asses if the measured value is far from a certified value to prevent systematic errors of measurements. The trueness is assessed for the ICP-OES method based on certified materials (Table B1). To address the comment from the reviewer, we have provided measured values on soil reference materials with their corresponding certified values (Table B1). Once the trueness of ICP-OES measurements was verified, a calibration was established between raw pXRF measurements and ICP-OES measurements on samples from a similar matrix (here on 144 samples of the Yedoma domain, i.e., 11% of the dataset) in order to correct for systematic error from the pXRF method.

**RC7:** Table 2. The disagreement of Al, Fe, Ca, K and Sr strongly exceed any uncertainties. The reliability of pXRF measurements is in question.

**AR:** We understand that Table 2 was confusing because the mean concentrations for both methods were not calculated based on the same number of samples. The purpose of Table 2 is to assess precision (i.e., random error) of both methods comparing relative standard deviations (%). To address the comment of the reviewer, we have revised Table 2 to provide the standard deviation on the same set of samples (n=3 for both methods) and we added the coefficient of variations (CV, expressed in %). The precision on the pXRF method was also assessed on a larger set of samples (n=20), and this is now presented in a new table in Appendix E. The text has been revised accordingly (L 238-244 in revised manuscript):

*"Given the influence of sample matrix on pXRF measurements, the precision of the pXRF method was also evaluated based on three to five repetitions on 20 individual samples and on average 2.6 times larger than based on three repetitions on three samples (Appendix E). The coefficient of variation was 20% smaller for Al but 6.5 times larger for Ca based on 20 samples. To ensure a cautious evaluation of the dataset, we decided to report the precision on the data in Fig. 5 based on the precisions from Table 2 for ICP-OES measurements, and from Appendix E for pXRF measurements to use the largest set of sample with precision data available."*

*Table 2: Precision of the element concentrations expressed as pooled standard deviations (i.e., two pooled standard deviations, expressed in mg kg$^{-1}$) for the 10 elements considered. The values are based on three repetitions of three identical samples for both ICP-OES and raw concentrations from pXRF method. The coefficient of variation (CV; expressed in %), defined as the ratio of the standard deviation to the mean is also provided.*

| | | Si | Al | Fe | Ca | K | Ti | Mn | Zn | Sr | Zr |
|---|---|---|---|---|---|---|---|---|---|---|---|
| ICP-OES | ±2SD$_{pooled}$ (mg kg$^{-1}$) | ± 1858 | ± 417 | ± 275 | ± 758 | ± 510 | ± 55 | ± 8.5 | ± 6.8 | ± 3.0 | ± 25 |
| | CV (%) | 0.29 | 0.43 | 0.50 | 0.99 | 2.01 | 0.71 | 0.85 | 4.31 | 1.05 | 4.21 |
| pXRF | ±2SD$_{pooled}$ (mg kg$^{-1}$) | ± 5887 | ± 3830 | ± 401 | ± 627 | ± 386 | ± 210 | ± 27 | ± 8.2 | ± 5.17 | ± 11.3 |
| | CV (%) | 1.15 | 3.75 | 0.74 | 0.84 | 1.51 | 2.88 | 2.91 | 5.72 | 1.97 | 2.04 |

*Appendix E: Precision of pXRF method on the element concentrations expressed pooled standard deviations (i.e., two pooled standard deviations, expressed in mg kg$^{-1}$) for the 10 elements considered. The values are based on a subset of Yedoma and Alas deposit samples with three repetitions of 20 samples. The coefficient of variation (CV; expressed in %), defined as the ratio of the standard deviation to the mean is available. The error bars on Fig. 5 are based on the following standard deviations for pXRF method.*

| | | Si | Al | Fe | Ca | K | Ti | Mn | Zn | Sr | Zr |
|---|---|---|---|---|---|---|---|---|---|---|---|
| pXRF | ±2SD$_{pooled}$ (mg kg$^{-1}$) | ± 3675 | ± 4107 | ± 2288 | ± 1066 | ± 1084 | ± 88.5 | ± 87 | ± 0.587 | ± 15.88 | ± 28.2 |
| | CV (%) | 2.14 | 3.07 | 3.44 | 5.46 | 2.77 | 3.62 | 8.74 | 7.78 | 4.09 | 4.93 |

**RC8:** The reasons of presenting correlations in Fig. 5 are unclear. Ca exhibits the highest correlation between two methods yet it is strongly (by a factor of 3.9) underestimated by pXRF (Table 2).

**AR:** This comment from the reviewer is related to the methodology and to his comment RC 7. The reason for presenting Figure 5 is the need to correct raw pXRF measurements (see response to comment RC 7). Figure 5 shows the correlation between raw pXRF measurements and accurate ICP-OES measurements. Correlations from Fig. 5 are essential because if pXRF and ICP-OES method were not correlated ($R^2 < 0.5$), the linear regression used to correct raw pXRF concentrations would not

correct the systematic error and corrected pXRF concentration would be biased. This is also the reason why we have chosen to present only mineral elements with reliable R² (R² > 0.7).

Table 2 shows the precision of measurements using ICP-OES and pXRF methods, and has been revised according to comment RC7 from the reviewer (see response to comment RC7).

**Specific comments**

**RC9:** L51-52 "associated to minerals" is unclear term. Does it imply stored in mineral (not peat) soils? Adsorbed onto /incorporated into mineral (such as Fe hydroxides or clays)?

**AR:** According to the comment of the reviewer, we have revised the manuscript (L 52 in the revised manuscript) as follows: "[…] *OC is associated to minerals through various mechanisms, detailed below*". The manuscript specifies the nature of those associations (i.e. aggregation, complexation, adsorption) on L 56-60: "*Mineral protection of OC includes aggregation, adsorption and/or complexation or co-precipitation processes (Kaiser and Guggenberger, 2003). These protection mechanisms involve i) clay minerals, Fe-, Al-, Mn-oxy-(hydr)oxides, or carbonates (aggregation); ii) clay minerals and Fe-, Al-, Mn-oxy-(hydr)oxides using polyvalent cation bridges such as $Fe^{3+}$, $Al^{3+}$, $Ca^{2+}$ or $Sr^{2+}$ (adsorption); or iii) $Fe^{3+}$, $Fe^{2+}$ and $Al^{3+}$ ions (complexation).*"

**RC10:** L123-130 and Fig.3: The vertical scale is missing / unclear Table 1: Total deposits depth and sampled depth are missing.

**AR:** A vertical scale with the mean thickness of ~20 meters and ~5.5 meters has been added to Fig. 3 (according to the response to RC1 about the thickness of the deposits).

**RC11:** L 153: One cannot use term 'soils' without providing soil sampling depth and soil classification. Were the samples subjected to < 2 mm sieving?

**AR:** We agree that the term "soils" is not appropriate to our deep sediments deposits (FAO defines "soil" has the natural medium for the growth of plant). We corrected this by replacing the term "*soils*" by "*deposits*".

To address the comment of the reviewer about sieving, we revised the MS L 195-196: "*The particle size distribution of these Yedoma domain deposits (described in reference papers from Table 1) is below 2 mm: therefore no sieving was necessary*."

**RC12:** L234 elements in ice wedges. It is true that their concentration is negligible compared to that in quartz and clays. However, the lability (rate of release from reservoir to the hydrological network) of Ca, K, Fe from the ice is million (or billion?) times faster than that of refractory minerals. One cannot neglect ice wedges in this context of elementary transport from permafrost to rivers and soils within the thawing scenario

**AR:** We thank the reviewer for raising this point related to ice wedges. Some studies have investigated the dissolved OC concentration as well as carbon and oxygen isotope composition of ice-wedges as archives of paleoclimate (e.g. Opel et al., 2018) but few data exist about mineral element concentration in ice-wedges (Fritz et al., 2011). Opel et al. (2018) states: the minor contributions of mineral particles and organic material during the formation of ice-wedges. In addition, Fritz et al. (2011) indicate that total ion content in ice is generally low with electrical conductivity of 212 µS/cm on average. The cation composition is dominated by $Na^+$ (58%) and followed by $Ca^{2+}$ with 30% on average (Fritz et al., 2011). Stocks in Na are not part of our assessment because pXRF cannot detect Na. Considering the Ca stock

in ice-wedges, we consider that this pool is limited and the flush of $Ca^{2+}$ from ice-wedge degradation may influence nutrient supply but only on a short time scale. Even if readily available, cations supply from ice wedges degradation is an ephemeral signal compared to long lasting solid-liquid interaction and mineral weathering upon thaw (see section 6.2). Nonetheless, we agree that it would be interesting in the future to include a more detailed assessment on the mineral elements stored in ice wedges, and we have revised the manuscript to include this comment (L 265-270 in revised manuscript):

"*Indeed, Opel et al. (2018) indicate the minor contribution of mineral particles during ice-wedge formation. Fritz et al. (2011) further indicate that ion concentrations in ice is generally low, dominated by $HCO3^-$ (55%) and Cl- (37%) for anions and $Na^+$ (58%) and $Ca^{2+}$ (30%) for cations. Note that a flush of highly labile mineral elements (e.g., $Na^+$, $Ca^{2+}$, $Cl^-$) locked inside ice-wedges may increase nutrient supply for a short time scale upon Yedoma deposits degradation compared to long-term solid-liquid interactions upon thaw.* "

**RC13:** L258-259 please provide the depth of these deposits, otherwise the surface area estimation is irrelevant

**AR:** This comment is related to comment RC1 and the manuscript has been corrected accordingly.

**RC14:** Table 3: Slope of dependences should be provided

**AR:** The pXRF device used here has its own geometry and therefore its own linear regressions. These equations cannot be reproduced with other pXRF devices (see response to comment RC 6). Therefore, providing the equation of the linear regression is not relevant for the reader since their pXRF device will require different regressions. Practically, other labs will need to produce a different regression for their pXRF device to correct their data. We have included an additional information in Figure 4 to clarify the method used to correct systematic error by pXRF (L 229 in revised manuscript).

The objective of our manuscript is to explain the data processing and the necessary steps to obtain reliable data using a pXRF device. We want to emphasize that raw pXRF data can be corrected with a linear regression specific to each pXRF device, and that this linear regression can be generated in-house by calibrating the pXRF method with another methodology such as ICP-OES measurements after alkaline digestion.

[Figure]

*Figure 4*: *(a) Comparison between two methods to assess mineral element concentrations in deposit samples: the inductively coupled plasma optical-emission spectrometry (ICP-OES) method (large blue arrow) is destructive and time-consuming (involving several steps), whereas the portable X-ray fluorescence (pXRF) method (dotted red arrows) is non-destructive and allows a fast and reliable determination of element concentrations on a large set of samples in a cost-effective way when a correction with a linear regression is applied. (b) The pXRF bias (i.e., systematic error) is corrected with a linear regression specific to each pXRF device. Portable XRF devices from each lab require their own linear regression to correct for accurate value. The linear regression is obtained from a selection of samples (here 11 % of the samples) analysed by both methods, pXRF and ICP-OES.*

We have further revised the manuscript (L 449-454 in revised manuscript): "*Linear regressions used to correct raw pXRF concentrations depend on internal geometry of the pXRF device used. This means that each pXRF device needs its own linear regression and that a single linear regression equation cannot be used with different pXRF devices. Moreover, pXRF measurements are also matrix-dependent. The matrix (i.e. organic content, bulk density) of a sample can affect pXRF-measured concentrations and therefore linear regressions must be calibrated with samples from the same matrix (here, we used a subset of 11% of samples from the Yedoma domain).*"

**RC15:** L290 This is relative uncertainty. Correctness of analysis is not discussed. Add slopes or 1:1 line to Fig. 5

**AR:** This comment relates to the previous comment RC 14. We have included an additional information in the text and in Figure 4 to clarify that raw pXRF measurement must be corrected for trueness with an established calibration, and to explain the method used to correct systematic error by pXRF (L 225). The 1:1 line in Figure 5 would only be relevant if both methods were characterized by a low systematic error, which is not the case for pXRF. The aim of the methodology is not to use raw pXRF data without any correction as explained L 220.

"*Raw pXRF concentrations cannot be used for absolute quantification if not corrected with a reliable and accurate method. Here, only pXRF concentration values corrected using a well-defined regression were used (Sect. 4.1).*"

**RC16:** L317-318: Table 3 does not allow to obtain reliable element concentrations; a linear equation with slope and intercept should be presented.

**AR:** This comment relates to the previous comment RC14 and RC15. We clarified in the revised manuscript (L 220) that *"Raw pXRF concentrations cannot be used for absolute quantification if not corrected with a reliable and accurate method. Here, only pXRF concentration values corrected using a well-defined regression were used (Sect. 4.1)."*

The pXRF device used here has its own geometry and therefore its own linear regressions. These equations cannot be reproduced with other pXRF devices (see response to comment RC14). Therefore, providing the equation of the linear regression is not relevant for the reader since their pXRF device will require different regressions. Practically, other labs will need to produce a different regression for their pXRF device to correct their data. We have included an additional information in Figure 4 and lines 449-455 in new Sect. 5) to clarify the method used to correct systematic error by pXRF.

**RC17:** L375-377: Depth of sediments is absolutely needed, otherwise the vulnerability cannot be assessed.

**AR:** According to the comment of the reviewer, the information about depth has been added (L 430 in revised manuscript): *"Absolute stock estimates (in Gt) allow direct comparison between Yedoma and Alas deposits, despite their different thickness (19.6 and 5.5 m, respectively)"*.

**RC18:** L420-242: What about other highly labile elements such as Li, B, Mg, Na, K, Sr, Ba?

**AR:** We agree with the reviewer that other elements would also be interesting to assess. The aim of this study is to perform a first assessment for selected relevant mineral elements regarding OC degradation/stabilization and mineral weathering and nutrient supply. We have included in the manuscript (L 534-536 in revised manuscript) that *"The YMCA dataset is a first step needed in order to evaluate the impact of widespread rapid permafrost thaw through thermokarst processes (Turetsky et al., 2020) on the mineral element concentrations in the deposits and the potential implications for OC and mineral nutrient supply".* We have also included a section with the limitations of the pXRF method (new section 5, L 448): *"In these deposits, the concentrations of some elements (Mg, P, Cu, Ni, Ba) could not be assessed due to poor pXRF-ICP-OES correlation ($R^2 < 0.5$)."* And L454: *"For some other elements, pXRF measurements are not possible due to the low atomic mass of these elements (N, Na)."* The reasons why some elements can not be measured by pXRF are explained L207: *"Because ambient air annihilates fluorescence photons that do not have enough energy, low atomic mass elements from Na and lighter cannot be quantified by pXRF."*

**RC19:** L428. Ca solubility is improper term (unless Ca is considered as native metal). One can speak about CaCO3, CaSO4 solubility. Otherwise use the term "mobility" or "lability" Section 5.3.1 is too general and irrelevant to the main findings of this study. Instead, based on mineral composition of studied yedoma soils, one can attempt to estimate the capacity of soil to capture organic matter. One can also explore element: C ratio to reveal which elements are most important for OC storage (unlikely that it would be Si, Mn, Sr. . .).

**AR:** We agree that Ca solubility was an improper term. We revised the sentence as follows (L 483-484): *"The YMCA dataset allows investigating the change in concentration of soluble elements such as Ca upon thermokarst processes resulting from permafrost thaw, i.e., between Yedoma and Alas deposits".*

According to the comment of the reviewer, we have also revised section 6.1 of the manuscript to investigate the evolution of Fe to organic carbon ratio, and Al to organic carbon ratio upon thawing between Yedoma and Alas deposits (new Fig. 8). We have also included a new section related to the

weathering potential of the minerals identified in the Yedoma deposits (section 6.2) based on the new mineralogical data presented (Table 6).

**RC20:** L435: These are not elements that interact with OC but minerals. More specifically, surface sites of minerals (>AlOH2+, >FeOH2+, >CaOH2+) can adsorb negatively charged organic ligands.

**AR:** We agree with the reviewer that surface sites of minerals are involved in OC stabilization within organo-mineral associations (adsorption onto mineral surfaces). In addition, we would like to specify that some elements (e.g., Fe, Al, Ca) can directly interact with dissolved OC to form complexes (organo-metallic complexes) without involving mineral surface sites. This is mentioned in the manuscript L 59-60.

**RC21:** L459-460: The primary nutrients in this context are N and P

**AR:** Among the 10 elements included in the YMCA dataset, macro- (e.g., K, Ca) and micro-nutrients (e.g., Fe, Mn, Zn) vital for plants and microorganisms are included. The N stock estimation in permafrost is investigated in other studies as mentioned L 64: "[…]*the increasing knowledge on N stocks (Fouché et al., 2020; Fuchs et al., 2018, 2019; Hugelius et al., 2020)"*.

We have revised the manuscript (L 458) to clarify that "*In these deposits, the concentrations of some elements (Mg, P, Cu, Ni, Ba) could not be assessed due to poor pXRF-ICP-OES correlation ($R^2 < 0.5$). For some other elements, pXRF measurements are not possible due to the low atomic mass of these elements (N, Na)."* We have also revised the manuscript (L 534) to acknowledge that "*The YMCA dataset is a first step needed in order to evaluate the impact of widespread rapid permafrost thaw through thermokarst processes (Turetsky et al., 2020) on the mineral element concentrations in the deposits and the potential implications for OC and mineral nutrient supply."*

**RC22:** L480: depth is needed

**AR:** The information about depth has been added (L 543 in revised manuscript).

"*This study provides the first mineral element inventory of permafrost deposits focusing on the ice-rich Yedoma region, i.e., never thawed Yedoma deposits and previously thawed Alas deposits for a mean thickness of 19.6 m and 5.5 m, respectively.*"

**RC23:** L485-486: vertical distribution is not shown in the paper; why it is presented in the Conclusions?

**AR:** We agree that the term "vertical distribution" was confusing. This sentence was revised as follows (L545 in revised manuscript):"*Based on the YMCA dataset, the total concentrations of 10 mineral elements in Yedoma domain deposits have been quantified in 75 different profiles."*

**RC24:** L490-493 Compared to fully instantaneous mobility of elements from ice wedges, the amount of elements that can be released from minerals such quartz or clays might be negligible at the time scale of decades.

**AR:** This comment refers to comment RC12 of the reviewer. We have revised the manuscript to include this point (L 265-270): "*Indeed, Opel et al. (2018) indicate the minor contribution of mineral particles during ice-wedge formation. Fritz et al. (2011) further indicate that ion concentrations in ice is generally low, dominated by $HCO_3^-$ (55%) and $Cl^-$ (37%) for anions and $Na^+$ (58%) and $Ca^{2+}$ (30%) for cations. Note that a flush of highly labile mineral elements (e.g., $Na^+$, $Ca^{2+}$, $Cl^-$) locked inside ice-wedges may increase nutrient supply for a short time scale upon Yedoma deposits degradation compared to long-term solid-liquid interactions upon thaw"*. In contrast with the ephemeral cations released from ice wedges

degradation, we have also included the fact that long lasting solid-liquid interaction and mineral weathering upon thaw (depending on the mineralogy of the deposit; new Table 6) may contribute to release mineral elements (see section 6.2).

**RC25:** The term 'stock' is highly confusing: in Fig B1 it is in Gt, but in Table D1 it is in kg/m3. These are totally different units.

**AR:** Following the reviewer's advice, we revised the manuscript using the term "mineral element density" (L 432 and Appendix H). The term stock (in Gt) is used for total mass (density x volume) of the minerals while mineral element density (in kg m$^{-3}$) is defined as mass of the mineral element divided by volume.

**RC26:** Fig. B1: L510+513 contradict to L515-516

**AR:** We agree the formulation was confusing. The caption has been revised as follows (L 597-600) to avoid confusion: *"This bootstrapping technique is used due to the non-normal distribution of the parameters. We used sampling with replacement, which means that after each step of the random draw from the original sample, we put the observation back before the following step. The process is done with 10 000 steps from which a stock density distribution is obtained. To estimate mineral elements stocks, we use the arithmetic mean and standard deviation assuming normality of the stock estimate distribution (Strauss et al., 2013)".*

Appendix A

**Table A1**: *Studied locations from the Yedoma domain with associated labels, total number of sampled profiles, number of samples analyzed with portable X-ray Fluorescence (pXRF), inductively coupled plasma optical emission spectrometry (ICP-OES) after alkaline fusion and X-ray diffraction method (XRD). A simplified geomorphological description of each sampling location is presented (detailed information is provided in the reference papers cited in Table 1). The site numbers (Site Nb) 1-17 are from Siberia, and 18-22 are from Alaska.*

| Site Nb | Site Name | Label | Total sampled profiles | Samples analyzed with pXRF method | Samples analyzed with ICP-OES method | Samples characterized with XRD method | Geomorphology of sampling location |
|---|---|---|---|---|---|---|---|
| 1 | Cape Mamontov Klyk | Mak | 3 | 80 | - | - | Sedimentary coastal plain |
| 2 | Nagym (Ebe Sise Island) | Nag | 3 | 29 | - | - | Cliff section with thermokarst mounds |
| 3 | Khardang Island | Kha | 1 | 31 | 1 | - | Cliff section with thermokarst mounds |
| 4 | Kurungnakh Island | Bkh, KUR | 4 | 143 | 2 | - | Fragment of a broad foreland plain north of the Chekanovsky Ridge |
| 5 | Sobo Sise Island | Sob | 4 | 58 | 58 | 13 | Yedoma uplands but also features permafrost degradation landforms (thermokarst lakes, drained thaw lake basin, and thermo-erosional gullies) |
| 6 | Bykovsky Peninsula | Mkh, BYK | 7 | 150 | 2 | - | Remnants of an accumulation plain |
| 7 | Muostakh Island | Muo | 1 | 11 | - | - | Remnants of an accumulation plain |
| 8 | Buor Khaya Peninsula | Buo | 5 | 80 | 44 | 10 | Coastal lowlands - late Pleistocene accumulation plains |
| 9 | Stolbovoy Island | Sto | 4 | 16 | 1 | - | Step-like cryoplanation terraces with several levels |
| 10 | Belkovsky Island | Bel | 2 | 12 | - | - | Step-like cryoplanation terraces with several levels |
| 11 | Kotel'ny Island | KyS | 1 | 10 | - | - | Step-like cryoplanation terraces with several levels |
| 12 | Bunge Land | Bun | 1 | 8 | - | - | More-or-less homogeneous flat sandy plain |
| 13 | Bol'shoy Lyakhovsky Island | TZ, R, L | 15 | 150 | 3 | - | Gradually sloping terrain intersected by rivers and thermo-erosional valleys |
| 14 | Oyogos Yar coast | Oy | 2 | 50 | 1 | - | Very gently inclined step-like surface of the Yana-Indigirka Lowland |
| 15 | Kytalyk | KY, KH | 3 | 50 | 4 | 4 | Recent and sub-recent floodplains, Yedoma and Alas |
| 16 | Duvanny Yar | DY | 6 | 143 | 5 | - | Hills dissected by deep thermos-erosional valleys and thermokarst depressions |
| 17 | Yukechi | Yuk-Yul | 4 | 87 | 2 | - | Yedoma uplands and drained alas basins, indicating active thermokarst processes |
| 18 | Kitluk | Kit | 2 | 45 | 2 | - | Tundra-covered coastal plain |
| 19 | Baldwin Peninsula | Bal | 4 | 70 | 1 | - | Sequence of marine, fluvial and glaciogenic sediments, which are well exposed along coastal bluffs and in some regions covered by loess-like deposits |
| 20 | Colville | Col | 1 | 23 | 8 | 3 | High exposure along the Colville River |
| 21 | Itkillik | Itk, It | 1 | 22 | 10 | 10 | High exposure along the lower Itkillik River formed by active river erosion of a large remnant of originally gently undulating yedoma terrain |
| 22 | Vault Creek Tunnel | FAI | 1 | 24 | - | - | Permafrost tunnel about 40 m deep and 220 m long on north facing slope |
| TOTAL | 22 | | 75 | 1292 | 144 | 40 | |

Appendix G

*Fig. G1*: *X-ray diffractograms of Yedoma, Alas and fluvial deposits in Siberia and Alaska (Q, quartz; Pl, Plagioclase; Ch, Chlorite; M, Mica; I, Illite; K, Kaolinite; D, Dolomite; Ca, Calcite; Sm, Smectite). Each diffractogram is labelled according to label code from Table 1 and samples depth is specified next to each label. (a) Sobo Sise profiles, (b) Buo-02 profile, (c) Buo-04 profile, (d) Buo-04-A-01 clay fraction (including the following treatments: K+ and Mg2+ saturation, ethylene glycol (eg) solvation and thermal treatments at 300 and 550°C), (e) Kytalyk profiles, (f) Colville profile and (g) Itkillik profile. Diffractograms include Yedoma (Sob T2-2, Sob T2-3, Buo-02, Buo-04, KY T1-1, Col, Itk) and Alas or fluvial deposits (Sob T2-5, Sob T2-6, KY T2-2). A summary of the mineral phases identified in each location is provided in Table 6.*

[Figure]

b)

[Figure]

Q

Pl

Ch M K

Buo-02-A-06

Buo-02-B-11

Buo-02-C-16

2-Theta - Scale

c)

[Figure]

Q

Pl

Ch M K

Buo-04-A-01

Buo-04-A-05

Buo-04-B-10

Buo-04-C-15

Buo-04-C-20

Buo-04-C-24

2-Theta - Scale

d)

Buo-04-A-01

2-Theta - Scale

e)

[Figure]

2-Theta - Scale

f)

[Figure]

g)

[Figure]

Cornelis, J.-T., Delvaux, B., Georg, R. B., Lucas, Y., Ranger, J. and Opfergelt, S.: Tracing the origin of dissolved silicon transferred from various soil-plant systems towards rivers: a review, Biogeosciences, 8(1), 89–112, https://doi.org/10.5194/bg-8-89-2011, 2011.

Fritz, M., Wetterich, S., Meyer, H., Schirrmeister, L., Lantuit, H. and Pollard, W. H.: Origin and characteristics of massive ground ice on Herschel Island (western Canadian Arctic) as revealed by stable water isotope and Hydrochemical signatures: Origin and Characteristics of Massive Ground Ice on Herschel Island, Permafr. Periglac. Process., 22(1), 26–38, https://doi.org/10.1002/ppp.714, 2011.

Goldich, S. S.: A Study in Rock-Weathering, J. Geol., 46(1), 17–58, https://doi.org/10.1086/624619, 1938.

Kahle, M., Kleber, M. and Jahn, R.: Retention of dissolved organic matter by phyllosilicate and soil clay fractions in relation to mineral properties, Org. Geochem., 35(3), 269–276, https://doi.org/10.1016/j.orggeochem.2003.11.008, 2004.

Murton, J. B., Goslar, T., Edwards, M. E., Bateman, M. D., Danilov, P. P., Savvinov, G. N., Gubin, S. V., Ghaleb, B., Haile, J., Kanevskiy, M., Lozhkin, A. V., Lupachev, A. V., Murton, D. K., Shur, Y., Tikhonov, A., Vasil'chuk, A. C., Vasil'chuk, Y. K. and Wolfe, S. A.: Palaeoenvironmental Interpretation of Yedoma Silt (Ice Complex) Deposition as Cold-Climate Loess, Duvanny Yar, Northeast Siberia: Palaeoenvironmental Interpretation of Yedoma Silt, Duvanny Yar, Permafr. Periglac. Process., 26(3), 208–288, https://doi.org/10.1002/ppp.1843, 2015.

National Research Centre for CRM: Institute of Geophysical and Geochemical Exploration component (GBW 07401- GBW 07404), Langfang, China, 1986.

Opel, T., Meyer, H., Wetterich, S., Laepple, T., Dereviagin, A. and Murton, J.: Ice wedges as archives of winter paleoclimate: A review, Permafr. Periglac. Process., 29(3), 199–209, https://doi.org/10.1002/ppp.1980, 2018.

Pewe, T. L. and Journaux, A.: Origin and character of loesslike silt in unglaciated south- central Yakutia, Siberia, USSR., US Geol. Surv. Prof. Pap., 1262, 1983.

Robert, M. and Tessier, D.: Méthode de préparation des argiles des sols pour des études minéralogiques, Méthode Préparation Argiles Sols Pour Études Minéralogiques, 25, 859–882, 1974.

Rouiller, J., Burtin, G. and Souchier, B.: La dispersion des sols dans l'analyse granulométrique. Méthode utilisant les résines échangeuses d'ions., , 14, 194–205, 1972.

Saidy, A. R., Smernik, R. J., Baldock, J. A., Kaiser, K., Sanderman, J. and Macdonald, L. M.: Effects of clay mineralogy and hydrous iron oxides on labile organic carbon stabilisation, Geoderma, 173–174, 104–110, https://doi.org/10.1016/j.geoderma.2011.12.030, 2012.

Schirrmeister, L., Froese, D., Tumskoy, V., Grosse, G. and Wetterich, S.: Yedoma: Late Pleistocene ice-rich syngenetic permafrost of Beringia, in Encyclopedia of Quaternary Science. 2nd edition, pp. 542–552, Elsevier, , 2013.

Strauss, J., Schirrmeister, L., Grosse, G., Wetterich, S., Ulrich, M., Herzschuh, U. and Hubberten, H.-W.: The deep permafrost carbon pool of the Yedoma region in Siberia and Alaska: Deep carbon of Siberia and Alaska, Geophys. Res. Lett., 40(23), 6165–6170, https://doi.org/10.1002/2013GL058088, 2013.

Tomirdiaro, S. V. and Chernen'kiy, O.: Cryogenic deposits of East Arctic and Sub Arctic, - SSSR Far-East-Sci. Cent., 196 pp., 1987.

Turetsky, M. R., Abbott, B. W., Jones, M. C., Anthony, K. W., Olefeldt, D., Schuur, E. A. G., Grosse, G., Kuhry, P., Hugelius, G., Koven, C., Lawrence, D. M., Gibson, C., Sannel, A. B. K. and McGuire, A. D.: Carbon release through abrupt permafrost thaw, Nat. Geosci., 13(2), 138–143, https://doi.org/10.1038/s41561-019-0526-0, 2020.

Wilson, M.: Weathering of the primary rock-forming minerals: processes, products and rates, Clay Miner., 39, 233–266, https://doi.org/10.1180/0009855043930133, 2004.

Wilson, S. A.: Data compilation for USGS reference material BHVO-2, Hawaian Basalt, US Geol. Surv. Open-File Rep., 2, 1997.